# Prioritized Model Experience Replay

**Muxi Tao** [1]  **Jiangtao Wen** [2]  **Yuxing Han** [1]

## Abstract

Model-based reinforcement learning (MBRL) improves sample efficiency by leveraging learned dynamics models, but often suffers from unstable training due to dynamics model learning mismatch: models are trained on data from historical policies while being queried under the continually updated current policy. This mismatch can cause policy-relevant local model error to remain large even as global prediction error decreases, leading to oscillatory updates. We present a finite-horizon performance analysis that decomposes the policy performance gap into global model error, policy-induced distribution shift, and historical policy mixture effects, showing that minimizing global error alone is insufficient for stable optimization. Motivated by this analysis, we propose Prioritized Model Experience Replay (PMER), a lightweight replay mechanism that prioritizes high-error transitions during dynamics model training. PMER implicitly emphasizes policy-relevant regions without explicit policy distance estimation and integrates seamlessly into Dyna-style MBRL frameworks. Experiments on MuJoCo benchmarks show improved stability, faster convergence, and higher sample efficiency.

## 1. Introduction

Model-based reinforcement learning (MBRL) improves sample efficiency by using learned dynamics models for policy optimization, but learning accurate models remains challenging due to limited data and model capacity (Abbas et al., 2020). As a result, policies can exploit model bias, leading to large gaps between predicted and true returns (Yu et al., 2020; Eysenbach et al., 2022; Rigter et al., 2023).

Project page available at: `https://pmer-icml2026.github.io/` [1]Shenzhen International Graduate School, Tsinghua University, Shenzhen, China [2]Department of Computer Science, New York University, New York, NY, United States. Correspondence to: Jiangtao Wen <jw9263@nyu.edu>.

*Proceedings of the 43rd International Conference on Machine Learning*, Seoul, South Korea. PMLR 306, 2026. Copyright 2026 by the author(s).

Despite advances such as model ensembles (Kurutach et al., 2018), trajectory-aware models (Janner et al., 2019; Seo et al., 2020), risk-sensitive objectives (Yu et al., 2020; Kidambi et al., 2020), and expressive architectures (Hafner et al., 2019; Bai et al., 2019; Zhang et al., 2023; Hafner et al., 2023), MBRL methods often exhibit unstable training, including oscillations and slow or failed convergence.

A key source of instability is dynamics model learning mismatch. In Dyna-style MBRL, dynamics models are trained on data collected by a mixture of historical policies but queried under the continually updated current policy, causing distributional mismatch. This leads to a divergence between global model error, measured over historical data, and local model error, measured under the current policy. While global error may decrease, local error can remain large and is often obscured during training, causing the model to appear accurate while producing highly misleading rollouts in policy-relevant regions.

Recent work such as Policy-adapted Dynamics Model Learning (PDML) (Wang et al., 2023) mitigates policy mixture effects by reweighting training data based on distances between historical and current policies. While effective, these methods require explicit policy distance estimation and additional bookkeeping, increasing computational overhead and potential instability. We observe that transitions with large dynamics model prediction error naturally correspond to policy-induced distribution shift, suggesting prediction error as a lightweight proxy for policy mismatch.

Motivated by this insight, we propose Prioritized Model Experience Replay (PMER), which selectively replays high-error transitions during dynamics model training. PMER implicitly emphasizes policy-relevant regions, reducing local model error while preserving global generalization, and integrates seamlessly into existing Dyna-style MBRL algorithms. We further present a finite-horizon performance analysis that decomposes the policy performance gap into global model error, policy-induced distribution shift, and historical policy mixture effects, explaining why minimizing global error alone may be insufficient. Experiments on MuJoCo benchmarks show that PMER improves training stability, accelerates convergence, and enhances sample efficiency across model-based reinforcement learning methods.

Our contributions are summarized as follows:

- We identify dynamics model learning mismatch as a crucial cause of instability in Dyna-style MBRL, and provide new insights into why error-based replay is effective: prediction error acts as a practical proxy for policy-induced mismatch, and replaying high-error transitions helps the dynamics model focus on underfitted, policy-relevant regions.

- We propose Prioritized Model Experience Replay (PMER), a lightweight replay mechanism that improves dynamics model learning by prioritizing high-error transitions without requiring explicit policy distance estimation. PMER can be seamlessly integrated into existing MBRL frameworks.

- We demonstrate through extensive experiments on MuJoCo benchmarks that PMER suppresses oscillatory learning, improves training stability, and significantly accelerates convergence across MBRL frameworks.

## 2. Background

### 2.1. Preliminaries

**Model-Based Reinforcement Learning.** A Markov Decision Process (MDP) is defined by the tuple $(\mathcal{S}, \mathcal{A}, T, r, \gamma)$, where $\mathcal{S}$ and $\mathcal{A}$ denote the state and action spaces, $T(s' \mid s, a)$ is the environment transition dynamics, $r(s, a)$ is the reward function, and $\gamma \in (0, 1)$ is the discount factor. The goal of reinforcement learning is to learn an optimal policy $\pi^*$ that maximizes the expected discounted return:

$$\pi^* = \arg\max_{\pi} \mathbb{E}\left[\sum_{t=0}^{\infty} \gamma^t r(s_t, a_t)\right], \qquad (1)$$

where $a_t \sim \pi(\cdot \mid s_t)$ and $s_{t+1} \sim T(\cdot \mid s_t, a_t)$. In model-based reinforcement learning (MBRL), a learned dynamics model $\hat{T}_\phi(s' \mid s, a)$, parameterized by $\phi$, is trained from data and used to support policy optimization.

**Dyna-Style Joint Objective.** Dyna-style algorithms do not rely exclusively on the learned model. Rather, they combine real environment interactions with short-horizon model rollouts to construct a truncated joint discounted return. Let $k$ denote the model rollout horizon. The objective can be written as

$$J(\pi) = \mathbb{E}_{\rho_T^\pi}\left[\sum_{t=0}^{k-1} \gamma^t r(s_t, a_t)\right] \\ + \gamma^k \mathbb{E}_{s_k \sim \rho_T^\pi} \mathbb{E}_{\rho_{\hat{T}_\phi}^\pi(\cdot \mid s_k)}\left[\sum_{t=0}^{\infty} \gamma^t r(\hat{s}_t, a_t)\right], \qquad (2)$$

where $\rho_T^\pi$ denotes the state–action distribution induced by policy $\pi$ under the true dynamics $T$, and $\rho_{\hat{T}_\phi}^\pi$ denotes the corresponding distribution under the learned dynamics $\hat{T}_\phi$.

### 2.2. Policy Mixture

As the policy is continuously updated during training, the environment buffer $\mathcal{D}_{\text{env}}$ inevitably contains transitions collected under both past and current policies, resulting in a policy mixture (Hazan et al., 2019). During dynamics model training, we consider a sequence of historical policies defined as $\Pi^n = \{\pi_1, \pi_2, \ldots, \pi_n\}$. Let $\rho^{\pi_i}(s, a)$ denote the state–action visitation distribution induced by policy $\pi_i$. We further associate each policy with a mixture weight $w_i^n$, forming the weight vector $\mathbf{w}^n = [w_1^n, \ldots, w_n^n]$. The mixture state–action distribution (Zhang et al., 2021) is

$$\rho^{\pi_{\text{mix}, n}}(s, a) = \sum_{i=1}^{n} w_i^n \rho^{\pi_i}(s, a). \qquad (3)$$

In Dyna-style dynamics model training, transitions are sampled uniformly at random from the replay buffer, which corresponds to assigning equal weights to all historical policies: $w_i^n = \frac{1}{n}, \quad \forall i \in \{1, \ldots, n\}$. The state–action distribution of the policy mixture simplifies to

$$\rho^{\pi_{\text{mix}, n}}(s, a) = \frac{1}{n} \sum_{i=1}^{n} \rho^{\pi_i}(s, a). \qquad (4)$$

### 2.3. Global and Local Model Error

We distinguish between global and local model error in model-based reinforcement learning. The global model error quantifies the prediction error of the learned dynamics model over the state–action distributions induced by all historical policies at iteration $n$. It is defined as $\epsilon_m = \max_t \mathbb{E}_{(s,a) \sim \rho^{\pi_{\text{mix}, n, t}}}\left[D_{\text{TV}}(T \parallel \hat{T}_\phi)\right]$, where $T$ denotes the true environment dynamics, $\hat{T}_\phi$ the learned model, and $\rho^{\pi_{\text{mix}, n, t}}$ the state–action distribution induced by the historical policy mixture at time step $t$. In practice, we estimate $\epsilon_m$ by monitoring the model's validation prediction loss on a held-out replay buffer after each training iteration, following the estimation used in (Janner et al., 2019).

The local model error characterizes the model prediction error under the state–action distribution induced by the current policy $\pi_n$. The local model error is defined as $\epsilon_{m'} = \max_t \mathbb{E}_{(s,a) \sim \rho^{\pi_n, t}}\left[D_{\text{TV}}(T \parallel \hat{T}_\phi)\right]$.

For small policy deviations, the local error can be approximated by $\epsilon_{m'}(\epsilon_\pi) \approx \epsilon_m + \epsilon_\pi \frac{d\epsilon_{m'}}{d\epsilon_\pi}$, where $\epsilon_\pi = \max_s D_{\text{TV}}(\pi_n(\cdot \mid s) \parallel \pi_{\text{mix}, n}(\cdot \mid s))$ measures the maximum total variation distance between the current policy and the policy distribution used to collect the training data. We denote

$$\delta_{\hat{T}_\phi} = \frac{d\epsilon_{m'}}{d\epsilon_\pi},$$

which characterizes the sensitivity of the local model error to policy-induced distribution shift.

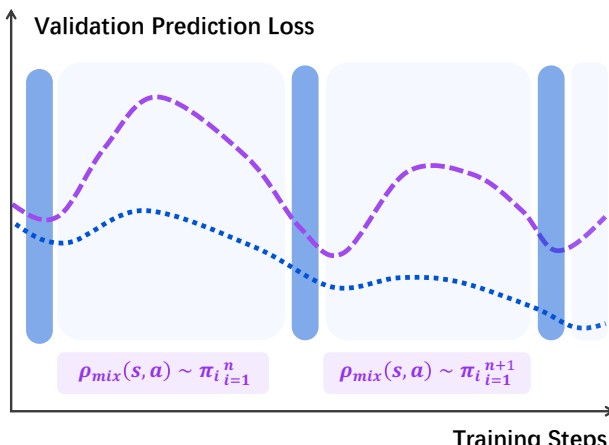

*Figure 1.* Policy-induced distribution shift causes persistent oscillations in local model error despite decreasing global error, motivating error-aware model learning.

## 2.4. Dynamics Model Learning Mismatch

During the training of MBRL algorithms, the policy is continuously updated, which inevitably induces a shift in the state–action visitation distribution. Such distributional shift caused by non-stationary policies is a major source of instability in reinforcement learning. In MBRL, this challenge becomes more pronounced. The dynamics model is trained not only on data collected by past policies, but is also expected to generalize to trajectories induced by newly updated policies. Meanwhile, an incompletely converged dynamics model is repeatedly used to roll out synthetic trajectories for policy optimization. This tight coupling between model learning and policy learning amplifies the adverse effects of distribution shift.

Consistent with prior theoretical analyses (Janner et al., 2019; Wang et al., 2023), we observe a non-negligible gap between the global model error $\epsilon_m$ and the local model error $\epsilon_{m'}$, as illustrated in Figure 1. When the dynamics model uniformly fits trajectories collected by all historical policies, its limited capacity and policy shift may prevent the local error $\epsilon_{m'}$ from decreasing effectively, which is detrimental to learning under the current policy. This mismatch leads to inaccurate model-generated samples that can misguide policy learning. Furthermore, policy updates driven by biased model rollouts tend to be unstable, often manifested as large oscillations in the training return curves. Such instability degrades sample efficiency and, in severe cases, prevents the policy from converging altogether.

## 3. Policy Performance Gap

In this section, we derive a performance gap bound that characterizes the discrepancy between policy performance in the learned dynamics model and in the true environment under dynamics model learning mismatch. The proposed theorem explicitly captures not only the effect of historical policy trajectories through a weighted policy mixture term, but also the impact of model capacity and environment-dependent model learning properties on policy optimization.

**Theorem 3.1** (Performance Gap under Model Learning Mismatch). *Let $J_T(\pi)$ denote the expected discounted return of policy $\pi$ in the true environment dynamics $T$, and let $J_{\hat{T}_\phi}(\pi)$ denote its performance under the learned dynamics model. Let $\gamma$ be the discount factor, $r_{\max}$ the maximum reward magnitude, and let $\mathcal{M}(w^n)$ denote the effect of the policy mixture induced by historical policy weights $w^n$ on model rollouts. At iteration $n$, under a short rollout horizon $k$, the performance gap between the real environment and the model rollout can be bounded as*

$$\left| J_T(\pi_n) - J_{\hat{T}_\phi}(\pi_n) \right| \leq 2\gamma r_{\max}\left(\epsilon_m + \mathcal{C}_{\hat{T}_\phi}\,\epsilon_\pi \right.$$
$$\left. + \mathcal{M}(w^n)\right) \quad (5)$$

*where the model capability $\mathcal{C}_{\hat{T}_\phi}$ is defined as*

$$\mathcal{C}_{\hat{T}_\phi} = \frac{1}{\gamma} + \delta_{\hat{T}_\phi} + \frac{1}{1 - \gamma L_s^{(k)}(\hat{T}_\phi)}.$$

*Proof.* See Appendix A. ∎

**Discussion.** The bound consists of three terms. First, to reduce the performance gap, the global model error $\epsilon_m$ should be minimized. This objective has been widely adopted in previous work, which aims to learn a dynamics model that fits the trajectories collected by all historical policies. However, we consistently observe that in many practical training runs, the global model error converges to a low value at an early stage of training, while policy performance and learning stability continue to improve only much later. This gap between early global error convergence and delayed policy stabilization motivates our study.

The second term shows that policy shift is amplified by the model capability, which is determined by $\delta_{\hat{T}_\phi}$ and $L_s^{(k)}(\hat{T}_\phi)$. Here, $\delta_{\hat{T}_\phi}$ denotes the upper bound on the sensitivity of the local error to policy shift. When the training data size of the model is sufficiently large, this coefficient can be treated as a relatively small constant (Janner et al., 2019). $L_s^{(k)}(\hat{T}_\phi)$ is the Dobrushin coefficient of the dynamics model, which characterizes the sensitivity of the model dynamics to state perturbations and controls the amplification of compounding errors in k-step rollouts (also referred to as model Lipschitz (Asadi et al., 2018)). We denote this coefficient as the

consistency coefficient. A detailed analysis and empirical validation for $L_s^{(k)}(\hat{T}_\phi)$ are provided in Appendix C.

The third term $\mathcal{M}(w^n)$ captures the effect of training the dynamics model on a mixture of historical policies, and corresponds to the standard uniform replay setting when $w_i^n = \frac{1}{n}$. In PDML (Wang et al., 2023), a weighting scheme over historical policies is proposed to assign importance weights to training samples collected under different policies, with the goal of mitigating the impact of policy mixture. Specifically, the weight assigned to the $i$-th historical policy at iteration $n$ is defined as

$$w_i^n = \frac{\xi_{\pi_i}^{-1}}{\sum_{m=1}^n \xi_{\pi_m}^{-1}}, \tag{6}$$

and $\xi_{\pi_i} = \mathbb{E}_{s \sim \rho_{\hat{T}_\phi}^{\pi_{\mathrm{mix},n}}(s)}[D_{\mathrm{TV}}(\pi_n(\cdot \mid s) \| \pi_i(\cdot \mid s))], \forall i \in [1, n-1]$. The distance between policies is estimated as $\xi_{\pi_i} = \frac{1}{K} \sum_{k=1}^K D_{\mathrm{TV}}(\pi_n(\cdot \mid s_k) \| \pi_i(\cdot \mid s_k))$. Here, $K$ denotes the total number of samples in the real sample buffer. PDML distributes weight $w_i^n$ to historical policy $\pi_i$ according to $\xi_{\pi_i}$, rather than using uniform sampling weights $w_i^n = \frac{1}{n}$. This reweighting successfully reduces the performance gap and leads to improved return performance. However, PDML relies heavily on fitting the dynamics model to the current policy distribution, which increases the computational overhead due to policy distance estimation and historical policy storage, and may weaken the global generalization of the model.

Theorem 3.1 suggests that reducing the global model error or the mixture-policy effect alone is not sufficient to align policy learning between the learned model and the real environment. In particular, the second term in Theorem 3.1 highlights the importance of reducing the sensitivity of the local model error to policy shift, as well as learning a dynamics model with better consistency that can effectively control the compounding error.

Our observation is that, since the local model error is typically larger than the global model error under policy shift, as suggested by prior analyses in Section 2.3 and 2.4 and consistently observed in practice, transitions with large dynamics model prediction error are more likely to originate from regions of the state–action space that are closer to the current policy, suggesting the prediction error as a lightweight proxy for policy mismatch. We empirically verify this intuition by showing that dynamics model prediction error correlates with policy-induced distribution shift (Appendix D).

This observation motivates an alternative strategy: rather than reweighting samples based on policy similarity, one can prioritize transitions according to their model prediction error. We can further influence the weights $w_i^n$ of historical policies in the mixture by repeatedly revisiting and relearning these high-error samples. In addition, by repeatedly revisiting high-error transitions, the dynamics model is en-

couraged to improve accuracy precisely in those regions that contribute most to the local performance gap and to become smoother at the local scale, implying an improved model consistency $L_s^{(k)}(\hat{T}_\phi)$.

## 4. Prioritized Model Experience Replay

In this section, we propose *Prioritized Model Experience Replay* (PMER), a novel mechanism for mitigating dynamics model learning mismatch by prioritizing high-error transitions and reweighting the influence of historical policies.

---

**Algorithm 1** Dynamics Model Training with PMER

---

1: **Input:** Prioritized model experience buffer $\mathcal{P} = \{(\tau_j, \psi_j)\}$, data buffer $\mathcal{D}$, batch size $l$, prior ratio $\rho$, decay coefficient $\lambda$, prior buffer capacity $K$
2: **for** each model training step **do**
3:     Sample transitions $\mathcal{T}_D = \{\tau(s, a, r, s')\}^{l \cdot (1-\rho)}$ uniformly from $\mathcal{D}$
4:     Sample prioritized $\mathcal{T}_P = \{\tau(s, a, r, s')\}^{l \cdot \rho}$ from $\mathcal{P}$ using $\beta$-sampling in Eq. 9
5:     Concatenate training batch $\mathcal{B} \leftarrow \mathcal{T}_D \cup \mathcal{T}_P$
6:     Compute prioritization weight $\{\psi_i\}_{\tau_i \in \mathcal{B}}$ as a byproduct of the training loss in Eq. 7
7:     Update dynamics model parameters using batch $\mathcal{B}$
8:     $\psi_j \leftarrow \lambda \psi_j$ for all $(\tau_j, \psi_j) \in \mathcal{P}$
9:     Assign weight $\psi_i$ to $\tau_i \in \mathcal{B}$ and rank the batch
10:     Insert the top $l \cdot \rho$ pairs $(\tau_i, \psi_i)$ from $\mathcal{B}$ into $\mathcal{P}$ and evict lowest-priority entries if $|\mathcal{P}| > K$
11: **end for**

---

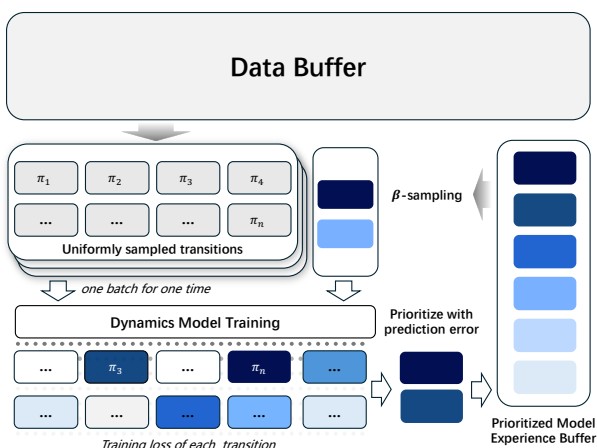

*Figure 2.* Illustration of the dynamics model training procedures enhanced by PMER.

Algorithm 1 and Figure 2 illustrate how we perform PMER during dynamics model training. Rather than reweighting all historical samples, PMER augments each dynamics model training batch with a fraction $\rho$ of high-error transitions

sampled from a prioritized buffer, while the remaining $(1 - \rho)$ fraction is sampled uniformly. The prioritized buffer is called Prioritized Model Experience Buffer.

For each replay sample, PMER assigns a priority score based on the discrepancy between the current model prediction and its corresponding supervised model-learning target. We denote this priority interface as

$$\psi_i = \|u_\theta(\mathbf{x}_i) - \mathbf{y}_i\|_2^2, \tag{7}$$

where $\mathbf{x}_i$ denotes the model input constructed from the replay sample, $\mathbf{y}_i$ denotes the corresponding supervised prediction target, and $u_\theta(\mathbf{x}_i)$ denotes the model prediction used to compute the priority. In our empirical implementation, we instantiate Eq. (7) with the Any-step Dynamics Model (ADM) used in ADMPO (Lin et al., 2024). For a $k_p$-step transition sequence collected by $\pi_n$ and indexed by $i$, the prioritization weight is computed as

$$\psi_i^n = \left\| \mu_\theta\left(\mathbf{s}_t^{(i)}, \mathbf{a}_{t:t+k_p-1}^{(i)}\right) - \begin{bmatrix} \mathbf{s}_{t+k_p}^{(i)} - \mathbf{s}_t^{(i)} \\ r_{t+k_p-1}^{(i)} \end{bmatrix} \right\|_2^2. \tag{8}$$

Here $\mu_\theta(\cdot)$ denotes the mean prediction of the ADM, whose target consists of the $k_p$-step state difference and the corresponding reward. All state and reward dimensions are normalized by their running standard deviation. Importantly, $\psi_i^n$ is used to control replay sampling and does not reweight the model training loss. Samples with larger $\psi_i^n$ are preferentially inserted into the prioritized buffer and later reused to augment subsequent training batches.

**Priority Weight Decay.** To avoid a small number of extreme-error transitions permanently dominating the prioritized buffer, we apply exponential decay to all stored priorities. Specifically, at every model update step, each stored priority is multiplied by a fixed decay coefficient $\lambda \in (0, 1)$. In all experiments, we set $\lambda = 0.995$. This gradually reduces the influence of stale high-error samples and ensures that the buffer reflects recent model weaknesses.

**$\beta$-Sampling.** The second mechanism follows the sampling strategy of Prioritized Experience Replay (PER) (Schaul et al., 2015). The sampling procedure is defined as follows. Let $\mathcal{I}_K = \{i_1, i_2, \ldots, i_K\}$ denote the selected index set, ordered such that $\psi_{i_1} \geq \psi_{i_2} \geq \cdots \geq \psi_{i_K}$. The sampling probability of the $r$-th ranked sample is defined as

$$p_\beta(i_r) = \frac{(K - r + 1)^\beta}{\sum_{j=1}^K (K - j + 1)^\beta}. \tag{9}$$

In practice, we find $\beta \in [2, 3]$ to provide a good trade-off between focusing on high-error transitions and maintaining sufficient diversity across tasks.

We further provide ablation on $\lambda$ and $\beta$ in Appendix E.

PMER induces an effective policy-mixture distribution for dynamics training, $\rho_{\text{train}}^n(s, a) = \sum_{i=1}^n \tilde{w}_i^n \rho^{\pi_i}(s, a)$, where

$$\tilde{w}_i^n = (1 - \rho) w_i^n + \rho \alpha_i^n. \tag{10}$$

Here $w_i^n$ denotes the baseline mixture weight from uniform replay, and $\alpha_i^n$ is the induced policy-level weight from prioritized replay, with $\sum_{i=1}^n \alpha_i^n = 1$. Let $\mathcal{P}_n$ be the prioritized buffer at iteration $n$ and let $p_\beta(\tau)$ be its rank-based $\beta$-sampling probability. Then $\alpha_i^n \triangleq \sum_{\tau \in \mathcal{P}_n \cap \mathcal{D}_i} p_\beta(\tau)$ represents the total prioritized sampling mass assigned to transitions collected by $\pi_i$.

**Discussion.** While being effective, PMER preserves the simplicity of the overall algorithmic framework and can be seamlessly integrated into model-based reinforcement learning algorithms (see Appendix F). By replaying high-error transitions, PMER implicitly reweights training data, which is equivalent to assigning larger effective weights to the corresponding policy mixture coefficients $w_i^n$, without explicit policy distance estimation or costly weight computation.

Moreover, PMER repeatedly exposes the dynamics model to more challenging regions of the state–action space, instead of overfitting to regions induced by recent policy updates. This partially mitigates training oscillations caused by excessive local optimization. As shown in our experiments, PMER facilitates more robust policy learning and reduces sensitivity to state perturbations.

In this work, we instantiate PMER with direct state and reward prediction error. In highly stochastic environments, prediction error may be dominated by intrinsic randomness rather than policy-induced mismatch, making prioritization less reliable. Extending the priority signal to latent or representation spaces is a promising direction for future work.

## 5. Experiments

### 5.1. Comparison with SOTA methods

In this section, we enhance the training of ADMPO with our method PMER and compare the resulting method against a set of representative baseline algorithms. ADMPO introduces concise yet effective modifications to MBPO by replacing the ensemble dynamics model with an Any-step Dynamics Model (ADM), which significantly reduces compounding errors while retaining a MBPO-style learning paradigm. For the model-free component, we compare against SAC as a strong representative baseline. As a primary policy-adaptive baseline closely related to our theoretical analysis, we include PDML-MBPO. In addition, we include several representative model-based baselines, including MBPO (Janner et al., 2019), AMPO (Shen et al., 2020), DDPPO (Li et al., 2022), and MACURA (Frauenknecht et al., 2024), which cover short-horizon model rollouts,

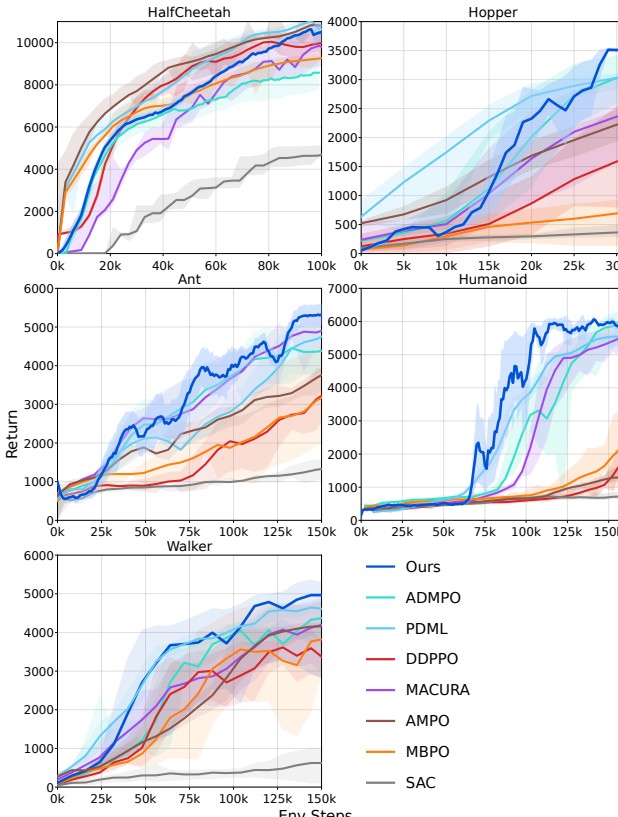

Figure 3. Performance curves of PMER and other baseline methods on five MuJoCo tasks. The solid lines illustrate the mean over five seeds and shaded regions represent the standard error.

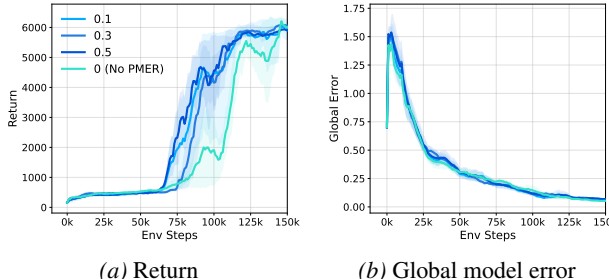

*(a)* Return      *(b)* Global model error

Figure 4. Ablation on the priority ratio $\rho$ on the Humanoid task. Left: policy return. Right: global model prediction error.

serve consistent gains; details are provided in Appendix F.

## 5.2. Ablation on the Priority Ratio

We study the sensitivity of PMER to the prioritized replay ratio $\rho$, which controls the fraction of high-error transitions injected into each dynamics model training batch. Figure 4 reports the learning curves and global model prediction errors under different $\rho$ on the complex Humanoid task. The results show that PMER consistently improves learning speed and stability across a wide range of $\rho$ values. Compared to $\rho = 0$ (no PMER), all prioritized variants enter high-return regimes substantially earlier, indicating that PMER does not rely on precise tuning of $\rho$ to be effective. While larger values of $\rho$ (e.g., $\rho = 0.5$) may introduce slightly higher variance during early training, the final performance remains comparable across all prioritized settings.

Importantly, the global model prediction error exhibits nearly identical convergence behavior across different $\rho$ values, suggesting that PMER does not outperform the baselines by simply reducing the global model prediction error. Instead, the performance gains mainly come from improving policy-relevant local model accuracy without sacrificing overall model fitting quality.

## 5.3. Sparse Return Validation

To expose training instability hidden by averaged validation returns, we introduce a sparse return validation experiment that periodically evaluates policies using a small number of real-environment rollouts. The results are shown in Figure 5.

The baseline method exhibits slow learning progress due to oscillatory updates induced by dynamics model learning mismatch. After policies reach high-return regions, policy exploitation and oscillations persist, which significantly degrade learning efficiency. In contrast, PMER improves policy robustness, with returns stabilizing after substantially fewer oscillations. On the Humanoid task, PMER reaches high-return regions earlier and approaches convergence shortly after the oscillatory phase between 75k and

adaptive rollout strategies, dynamics distillation, and uncertainty-aware conservative regularization. The comprehensive performance comparisons are shown in Figure 3 and detailed settings are provided in Appendix G.

We observe that our method achieves a significantly faster learning speed compared to its backbone baseline ADMPO, which clearly demonstrates the improvement in learning efficiency brought by PMER. Moreover, across all five evaluation environments, PMER consistently exhibits competitive learning efficiency among all baseline methods.

On the Hopper benchmark, our method surpasses a return of 3500 within approximately 30k environment steps, whereas most competing methods only approach this performance level around 50k steps. In the Humanoid environment, the PMER-enhanced method explores high-return regions at around 80k steps and converges to a return close to 6000 by approximately 100k steps. In contrast, the baseline ADMPO without PMER converges to a similar performance level only around 140k steps, while PDML-MBPO reaches comparable performance at approximately 150k steps. We also evaluate PMER on the standard MBPO backbone and ob-

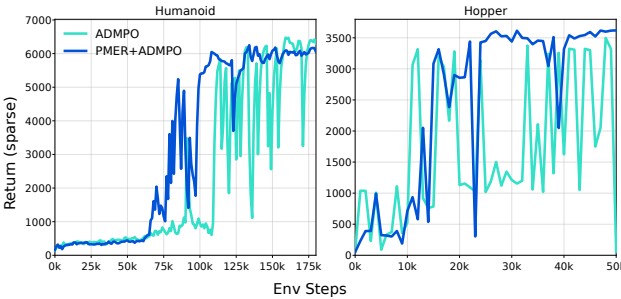

*Figure 5.* Policies are evaluated in the real environment every 1k training steps using 10 validation rollouts under the same random seed. The figure reports the raw validation returns revealing oscillatory behaviors during training. PMER leads to faster stabilization and reduced return oscillations compared to baseline methods.

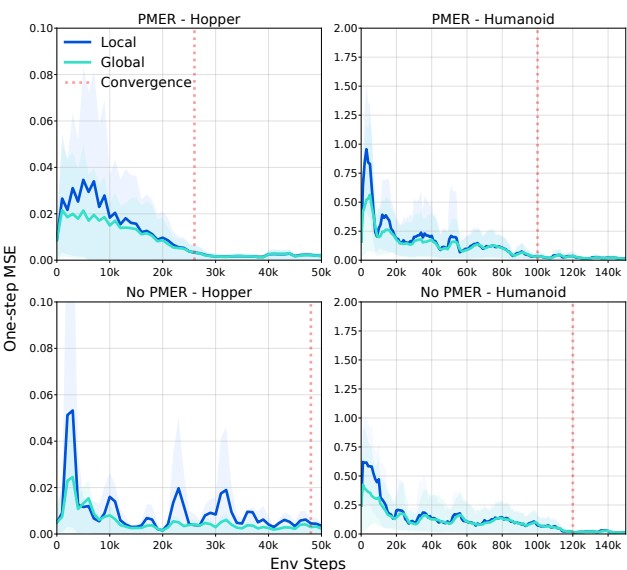

*Figure 6.* Comparison of global and local one-step prediction error with and without PMER on the Hopper and Humanoid tasks. PMER better aligns the reduction of global and local errors, mitigating the model learning mismatch and accelerating convergence.

100k steps. The baseline method continues to exhibit pronounced oscillations until around 150k steps before gradually stabilizing. These results indicate that PMER improves robustness and learning efficiency by enhancing dynamics model learning, without modifying the policy optimization procedure.

### 5.4. Global and Local Model Error Analysis

Figure 6 visualizes the evolution of one-step dynamics model prediction errors throughout training, comparing the cases with and without PMER. When PMER is applied, the global error and local error decrease in a smooth and synchronized manner. Notably on the Hopper task, around

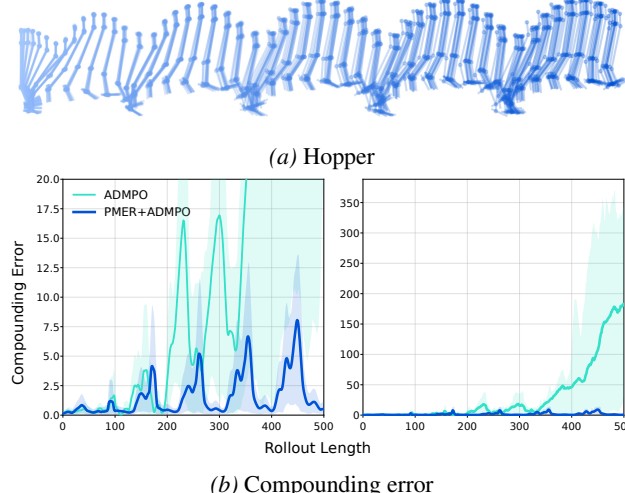

*(a)* Hopper

*(b)* Compounding error

*Figure 7.* (a) Using a converged policy, we collect 20 rollouts of length 500 in the real environment and overlay the trajectories of four key body states of the Hopper. (b) The compounding error refers to the accumulation of model prediction errors along a rollout. The PMER-enhanced method captures the periodic structure of the Hopper motion and exhibits improved compounding error control compared to the baseline method.

26k training steps, the local error converges to the same level as the global error. This alignment coincides with the convergence point observed in the policy return curve.

Without PMER, although the global error decreases to a relatively small value at an early stage of training, the local error continues to oscillate throughout the learning process. This behavior is consistent with persistent policy-induced distribution shift caused by dynamics model learning mismatch. As a result, the dynamics model generates inaccurate rollouts under updated policies, increasing the risk of policy exploitation. Such instability aligns with the oscillatory policy behavior observed in Figure 5.

Importantly, PMER operates solely on dynamics model learning, without modifying the model architecture. The results show that PMER effectively alleviates dynamics model learning mismatch in model-based reinforcement learning, which promotes a synchronized reduction of global and local errors, leading to faster and more stable convergence.

### 5.5. Comparison of Compounding Error

Figure 7 shows how the compounding error evolves as the rollout horizon increases. Both methods maintain reasonable error control over moderate horizons; however, PMER exhibits substantially improved stability and error control in ultra-long rollouts. Under PMER, the compounding error curve exhibits a clear periodic structure. In contrast, the baseline captures only the initial cycles before compounding error grows rapidly and becomes unstable. Notably,

the compounding error periodically drops at large horizons (around 200, 300, and 400 steps). These drops correspond to repeated visits to similar postures near the apex of the jumping motion. Higher uncertainty is instead observed during rapid landing and take-off phases.

This behavior is consistent with improved control of the model consistency factor $L_s^{(k)}(\hat{T}_\phi)$. By repeatedly incorporating high-error transitions, PMER encourages smoother dynamics predictions in high-bias regions, which aligns with a smaller effective consistency factor and reduced rollout amplification. Further discussion of the consistency factor is provided in Appendix C.

## 6. Related Work

**Model-Based Reinforcement Learning.** Model-based reinforcement learning (MBRL) aims to reduce the sample complexity of model-free reinforcement learning by learning an explicit dynamics model for policy optimization. Early approaches relied on linear dynamics model formulations (Parr et al., 2008), while probabilistic Gaussian process models were later introduced to capture uncertainty (Seeger, 2004; Grande et al., 2014). With the success of deep learning, neural-network-based dynamics models have become the dominant choice (Chua et al., 2018). Subsequent work focused on improving model accuracy through multi-step prediction (Asadi et al., 2018) or mitigating distribution mismatch via unsupervised or representation model learning (Berkenkamp et al., 2017; Zhang et al., 2019; Shen et al., 2020; Zhang et al., 2023). In contrast to approaches that modify model architectures or training objectives, PMER operates at the data replay level and biases model learning toward policy-relevant regions via error-aware sampling.

**Model usage and rollout strategies.** Beyond model accuracy, a substantial body of work investigates how learned dynamics models are used during policy optimization. Janner et al. (Janner et al., 2019) theoretically and empirically showed that short-horizon rollouts can effectively mitigate compounding error, motivating the MBPO framework. Building on this insight, bidirectional or adaptive rollout strategies were proposed to further control rollout bias (Lai et al., 2020; 2021). Other approaches leverage model uncertainty or disagreement to regulate when and how model rollouts are trusted (Pan et al., 2020; Yu et al., 2020; Wu et al., 2022). These methods primarily address model error at the model usage or policy optimization stage. In contrast, PMER focuses on the model learning stage by adjusting how training data are replayed, implicitly reducing policy-dependent prediction errors before rollouts are generated.

**Experience replay and prioritization.** Experience replay is a central component of off-policy reinforcement learning, enabling efficient reuse of past interactions (Lin, 1992; Mnih et al., 2015). Prioritized Experience Replay (PER) (Schaul et al., 2015) samples transitions according to temporal-difference (TD) error and has demonstrated strong empirical performance in value-based methods (Zhang & Sutton, 2017; Hessel et al., 2018; Oh et al., 2021). However, TD-error-based prioritization depends heavily on critic accuracy and can be unstable or less effective in policy-based or actor–critic settings (Yu, 2018; Zha et al., 2019; Peer et al., 2021; Saglam et al., 2023). In contrast, PMER applies prioritization exclusively to dynamics model learning and uses model prediction error, rather than value estimation error, as the prioritization signal. Model-augmented Prioritized Experience Replay (MaPER) (Oh et al., 2021) is a replay-prioritization method that augments PER with model-related auxiliary signals to improve Q-value learning; specifically, its priority combines TD error, reward prediction error, and transition prediction error using adaptive coefficients. In contrast, PMER applies prioritization only to dynamics model training, using a single dynamics-prediction-error signal motivated by the mismatch between the historical training distribution and the current-policy rollout distribution. Curious Replay (Kauvar et al., 2023) is motivated by rapid adaptation in non-stationary environments and prioritizes experiences based on model unfamiliarity and replay frequency. Prioritized Generative Replay (PGR) (Wang et al., 2024) also adopts an error-based relevance signal in a generative replay setting, where latent-space forward prediction error is used to condition a diffusion model to generate imagined transitions from past experience. These works provide strong empirical evidence that curiosity- or error-based replay can improve sample efficiency. In contrast, PMER focuses on the underlying mechanism and provides new insights into why error-based replay is effective and operationalizes this insight through a lightweight replay mechanism.

**Policy-adaptive dynamics model learning.** Recent work has studied dynamics model learning under evolving policies, showing that models trained on data from historical policies can incur large prediction errors under the current policy, leading to biased rollouts and unstable updates (Luo et al., 2018; Janner et al., 2019; Kidambi et al., 2020). To address this issue, several methods adapt model learning to policy-induced distribution shift. PDML reweights training samples according to the distance between historical policies and the current policy to mitigate policy mixture effects in dynamics fitting (Wang et al., 2023). Related ideas also appear in joint model–policy optimization frameworks, where the dynamics model and policy are co-optimized to reduce mismatch during training (Clavera et al., 2018; Rajeswaran et al., 2020; Eysenbach et al., 2022). More broadly, policy adaptivity has been explored via importance weighting and on-policy data emphasis in both model-based and model-

free settings (Schulman et al., 2015; 2017; Van Hasselt et al., 2019; Zha et al., 2019). While these approaches rely on explicit policy distance estimation or importance weighting, PMER provides an implicit alternative by using dynamics model prediction error as a proxy for policy-induced distribution shift, enabling seamless integration into existing MBRL frameworks.

## 7. Conclusion and Discussion

This paper identifies dynamics model learning mismatch as a central cause of instability in model-based reinforcement learning, showing that policy performance is governed by policy-dependent local model error rather than global prediction accuracy alone. Our analysis clarifies how distribution shift and historical policy mixtures amplify rollout bias and lead to oscillatory learning behavior.

Building on this insight, we propose Prioritized Model Experience Replay (PMER), a simple and general mechanism that prioritizes high-error transitions during dynamics model training. PMER requires no explicit policy distance estimation or model architectural changes and integrates seamlessly into existing MBRL frameworks. Across standard continuous-control benchmarks, PMER consistently improves training stability and convergence while preserving global model accuracy. These results establish error-aware replay as an effective principle for stabilizing and scaling model-based reinforcement learning.

Despite these advances, PMER operates purely at the data replay level and thus relies on the capacity of the underlying dynamics model. In highly stochastic environments, prediction error may be dominated by intrinsic randomness rather than reducible model bias, making it less reliable for identifying policy-relevant regions. Extending PMER to discrete, high-dimensional, and real-world domains is an important direction for future work.

## Acknowledgements

We thank the anonymous reviewers for their constructive feedback. The first author is deeply grateful to his late grandfather, who passed away during the completion of this work, for his lifelong care, encouragement, and support.

## Impact Statement

This paper presents work whose goal is to advance the field of Model-Based Reinforcement Learning. There are many potential societal consequences of our work, none of which we feel must be specifically highlighted here.

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

# A. Policy Performance Gap

In this appendix, we provide the proof of Theorem 3.1.

**Assumption A.1** (Local model error under policy shift). For each iteration $n$, we assume that the local dynamics model error under the current policy can be upper bounded by the global model error induced by the historical policy mixture and an additional policy-dependent mismatch term:

$$\epsilon_{m'}^{(n)} \leq \epsilon_m^{(n)} + \delta_{\hat{T}_\phi} \epsilon_\pi^{(n)}. \tag{11}$$

where $\epsilon_\pi^{(n)}$ measures the policy-induced distribution shift and $\delta_{\hat{T}_\phi} \geq 0$ is a sensitivity coefficient. This approximation follows the local-error analysis commonly used in MBPO-style bounds (Janner et al., 2019). For simplicity, we omit the iteration index $(n)$.

**Theorem A.2** (Performance Gap under Model Learning Mismatch). *Let $J_T(\pi)$ denote the expected discounted return of policy $\pi$ in the true environment dynamics $T$, and let $J_{\hat{T}_\phi}(\pi)$ denote its performance under the learned dynamics model. Let $\gamma$ be the discount factor, $r_{\max}$ the maximum reward magnitude, and let $\mathcal{M}(w^n)$ denote the effect of the policy mixture induced by historical policy weights $w^n$ on model rollouts. At iteration $n$, under a short rollout horizon $k$, the performance gap between the real environment and the model rollout can be bounded as*

$$\left| J_T(\pi_n) - J_{\hat{T}_\phi}(\pi_n) \right| \leq 2\gamma r_{\max} \left( \epsilon_m + \mathcal{C}_{\hat{T}_\phi} \epsilon_\pi + \mathcal{M}(w^n) \right). \tag{12}$$

*where the model capability $\mathcal{C}_{\hat{T}_\phi}$ is defined as*

$$\mathcal{C}_{\hat{T}_\phi} = \frac{1}{\gamma} + \delta_{\hat{T}_\phi} + \frac{1}{1 - \gamma L_s^{(k)}(\hat{T}_\phi)}.$$

*Proof.* Let $\epsilon_{m'}$ denote the local error:

$$\epsilon_{m'} = \max_t \mathbb{E}_{(s,a) \sim \rho^{\pi_n, t}} \left[ D_{\text{TV}} \left( T(\cdot \mid s, a) \| \hat{T}_\phi(\cdot \mid s, a) \right) \right], \tag{13}$$

Applying the first term in Lemma B.1,

$$2\gamma r_{\max} \mathbb{E}_{(s,a) \sim \rho_T^\pi} \left[ D_{\text{TV}} \left( T(\cdot \mid s, a) \| \hat{T}(\cdot \mid s, a) \right) \right], \tag{14}$$

we note that the state–action visitation distribution is induced by the current policy $\pi$. In this case, we can upper bound the model error under the current occupancy by the maximum local model error $\epsilon_{m'}$, i.e.,

$$\mathbb{E}_{(s,a) \sim \rho_T^\pi} \left[ D_{\text{TV}} \left( T(\cdot \mid s, a) \| \hat{T}(\cdot \mid s, a) \right) \right] \leq \epsilon_{m'}. \tag{15}$$

Therefore, Term (I) in Lemma B.1 can be rewritten as

$$2\gamma r_{\max} \mathbb{E}_{(s,a) \sim \rho_T^\pi} \left[ D_{\text{TV}} \left( T(\cdot \mid s, a) \| \hat{T}(\cdot \mid s, a) \right) \right] \leq 2\gamma r_{\max} \epsilon_{m'}. \tag{16}$$

Term (II) in Lemma B.1 can be simplified into an operator controlled by the weights $w^n$,:

$$\mathcal{M}(w^n)_{\left( \{\xi_{\rho_i}\}, \{\xi_{\pi_i}\} \right)} = r_{\max} \sum_{i=0}^{k} w_i^n \left( \frac{1}{2} \text{Vol}(\mathcal{S}) \xi_{\rho_i} + \frac{1}{\gamma} \xi_{\pi_i} \right). \tag{17}$$

This expression indicates that, to reduce the performance gap when leveraging a sequence of historical policies $\Pi^n = \{\pi_0, \ldots, \pi_n\}$, one should not assign uniform weights to all historical policies. Instead, each historical policy $\pi_i$ should be assigned an adaptive weight $w_i^n$, and the weight should be negatively correlated with the distribution shifts $\xi_{\rho_i}$ and $\xi_{\pi_i}$.

Finally, we rewrite the weighted sum term as

$$2\gamma r_{\max}\, \mathcal{M}(w^n)\big(\{\xi_{\rho_i}\}, \{\xi_{\pi_i}\}\big) = r_{\max} \sum_{i=0}^{n} w_i^n \Big(\gamma\, \mathrm{Vol}(\mathcal{S})\, \xi_{\rho_i} + 2\, \xi_{\pi_i}\Big). \tag{18}$$

Applying Term (III) in Lemma B.1, we first apply Lemma B.2 (with $\pi_{\mathrm{D}} \leftarrow \pi_{\mathrm{mix},n}$) to upper bound. Specifically,

$$\begin{aligned}
&2 r_{\max}\, D_{\mathrm{TV}}\Big(\rho_{\hat{T}_\phi}^{\pi_{\mathrm{mix},n}}(s,a) \,\|\, \rho_{\hat{T}_\phi}^{\pi}(s,a)\Big) \\
&\leq 2 r_{\max}\Bigg(D_{\mathrm{TV}}\Big(v_{\hat{T}_\phi}^{\pi}(s) \,\|\, v_{\hat{T}_\phi}^{\pi_{\mathrm{mix},n}}(s)\Big) + \mathbb{E}_{s \sim v_{\hat{T}_\phi}^{\pi}}\Big[D_{\mathrm{TV}}(\pi(\cdot \mid s) \,\|\, \pi_{\mathrm{mix},n}(\cdot \mid s))\Big]\Bigg).
\end{aligned} \tag{19}$$

From our earlier definition, we assume a uniform bound on the policy deviation:

$$\max_s\ D_{\mathrm{TV}}(\pi(\cdot \mid s) \,\|\, \pi_{\mathrm{mix},n}(\cdot \mid s)) \leq \epsilon_\pi. \tag{20}$$

Thus, the second term can be upper bounded by $\epsilon_\pi$, yielding

$$2 r_{\max}\, D_{\mathrm{TV}}\Big(\rho_{\hat{T}_\phi}^{\pi_{\mathrm{mix},n}}(s,a) \,\|\, \rho_{\hat{T}_\phi}^{\pi}(s,a)\Big) \leq 2 r_{\max}\Big(D_{\mathrm{TV}}\Big(v_{\hat{T}_\phi}^{\pi} \,\|\, v_{\hat{T}_\phi}^{\pi_{\mathrm{mix},n}}\Big) + \epsilon_\pi\Big). \tag{21}$$

Next, assuming the dynamics model induces a TV contraction with constant $L_s^{(k)}(\hat{T}_\phi)$, we apply Lemma B.5 to control the visitation mismatch:

$$D_{\mathrm{TV}}\Big(v_{\hat{T}_\phi}^{\pi} \,\|\, v_{\hat{T}_\phi}^{\pi_{\mathrm{mix},n}}\Big) \leq \frac{\gamma}{1 - \gamma L_s^{(k)}(\hat{T}_\phi)}\, \mathbb{E}_{s \sim v_{\hat{T}_\phi}^{\pi}}[D_{\mathrm{TV}}(\pi(\cdot \mid s) \,\|\, \pi_{\mathrm{mix},n}(\cdot \mid s))]. \tag{22}$$

Using $\epsilon_\pi$ as an upper bound on the per-state policy TV distance, we obtain

$$D_{\mathrm{TV}}\Big(v_{\hat{T}_\phi}^{\pi} \,\|\, v_{\hat{T}_\phi}^{\pi_{\mathrm{mix},n}}\Big) \leq \frac{\gamma}{1 - \gamma L_s^{(k)}(\hat{T}_\phi)}\, \epsilon_\pi. \tag{23}$$

Combining the above inequalities gives the desired bound:

$$2 r_{\max}\, D_{\mathrm{TV}}\Big(\rho_{\hat{T}_\phi}^{\pi_{\mathrm{mix},n}}(s,a) \,\|\, \rho_{\hat{T}_\phi}^{\pi}(s,a)\Big) \leq 2 r_{\max}\left(\frac{\gamma}{1 - \gamma L_s^{(k)}(\hat{T}_\phi)} + 1\right) \epsilon_\pi. \tag{24}$$

Summing the three bounds yields

$$2\gamma r_{\max}\epsilon_{m'}\ +\ 2 r_{\max}\left(\frac{\gamma}{1 - \gamma L_s^{(k)}(\hat{T}_\phi)} + 1\right) \epsilon_\pi\ +\ 2\gamma r_{\max}\, \mathcal{M}(w^n)\big(\{\xi_{\rho_i}\}, \{\xi_{\pi_i}\}\big). \tag{25}$$

Finally, applying Assumption A.1, we relate the local model error to the global error and the policy shift. Substituting this bound into the Term 1 contribution $2\gamma r_{\max}\epsilon_{m'}$ yields

$$2\gamma r_{\max}\epsilon_{m'}\ \leq\ 2\gamma r_{\max}\epsilon_m + 2\gamma r_{\max}\delta_{\hat{T}_\phi}\epsilon_\pi. \tag{26}$$

Collecting the $\epsilon_\pi$ terms completes the proof. $\square$

## B. Useful Lemmas

**Lemma B.1** (Performance gap bound under historical policy mixture). *Let $T$ be the true dynamics and $\hat{T}_\phi$ a learned model. Let the historical mixture policy at iteration $n$ be*

$$\pi_{\mathrm{mix},n} = \sum_{i=0}^{n} w_i^n \pi_i, \tag{27}$$

*with weights $w^n$. Assume bounded rewards $|r(s,a)| \le r_{\max}$ and discount factor $\gamma \in (0,1)$. Define*

$$\xi_{\rho_i} = D_{\mathrm{TV}}\Big(\rho_T^\pi(s,a) \,\big\|\, \rho_T^{\pi_i}(s,a)\Big), \qquad \xi_{\pi_i} = \mathbb{E}_{s \sim v_{\hat{T}_\phi}^{\pi_{\mathrm{mix},n}}}\Big[D_{\mathrm{TV}}\big(\pi(\cdot|s) \,\big\|\, \pi_i(\cdot|s)\big)\Big], \tag{28}$$

*where $v_{\hat{T}_\phi}^{\pi_{\mathrm{mix},n}}$ denotes the state visitation distribution induced by $\pi_{\mathrm{mix},n}$ under $\hat{T}_\phi$. Then*

$$
\begin{aligned}
J(\pi,T) - J(\pi,\hat{T}_\phi) \ \le\ & \underbrace{2\gamma r_{\max} \, \mathbb{E}_{(s,a)\sim\rho_T^\pi}\Big[D_{\mathrm{TV}}\big(T(\cdot|s,a) \,\big\|\, \hat{T}_\phi(\cdot|s,a)\big)\Big]}_{\textbf{(I) On-policy model error}} \\
& + \underbrace{r_{\max} \sum_{i=0}^{n} w_i^n \Big(\gamma \operatorname{Vol}(\mathcal{S})\, \xi_{\rho_i} + 2\xi_{\pi_i}\Big)}_{\textbf{(II) Historical mixture term } \mathcal{M}(w^n)} \\
& + \underbrace{2 r_{\max}\, D_{\mathrm{TV}}\Big(\rho_{\hat{T}_\phi}^{\pi_{\mathrm{mix},n}}(s,a) \,\big\|\, \rho_{\hat{T}_\phi}^{\pi}(s,a)\Big)}_{\textbf{(III) Occupancy mismatch under } \hat{T}_\phi}.
\end{aligned}
\tag{29}
$$

*Proof.* The proof follows the decomposition and bounding steps in (Wang et al., 2023), with notation adapted to our setting. Introduce the mixture policy and decompose

$$
\begin{aligned}
J(\pi,T) - J(\pi,\hat{T}_\phi) &= \Big(J(\pi,T) - J(\pi_{\mathrm{mix},n},\hat{T}_\phi)\Big) + \Big(J(\pi_{\mathrm{mix},n},\hat{T}_\phi) - J(\pi,\hat{T}_\phi)\Big) \\
&\le \left| \int \Big(\rho_T^\pi(s,a) - \rho_{\hat{T}_\phi}^{\pi_{\mathrm{mix},n}}(s,a)\Big) r(s,a)\,ds\,da \right| \\
&\quad + \left| \int \Big(\rho_{\hat{T}_\phi}^{\pi_{\mathrm{mix},n}}(s,a) - \rho_{\hat{T}_\phi}^{\pi}(s,a)\Big) r(s,a)\,ds\,da \right|.
\end{aligned}
\tag{30}
$$

Using $|r| \le r_{\max}$ and the definition of total variation,

$$\left| \int \Big(\rho_{\hat{T}_\phi}^{\pi_{\mathrm{mix},n}}(s,a) - \rho_{\hat{T}_\phi}^{\pi}(s,a)\Big) r(s,a)\,ds\,da \right| \le 2 r_{\max}\, D_{\mathrm{TV}}\Big(\rho_{\hat{T}_\phi}^{\pi_{\mathrm{mix},n}}(s,a) \,\big\|\, \rho_{\hat{T}_\phi}^{\pi}(s,a)\Big). \tag{31}$$

For the remaining term, write $\rho(s,a) = v(s)\pi(a|s)$ and add/subtract $v_{\hat{T}_\phi}^{\pi_{\mathrm{mix},n}}(s)\pi(a|s)$:

$$
\begin{aligned}
& \left| \int \Big(\rho_T^\pi(s,a) - \rho_{\hat{T}_\phi}^{\pi_{\mathrm{mix},n}}(s,a)\Big) r(s,a)\,ds\,da \right| \\
&= \left| \int \Big(v_T^\pi(s)\pi(a|s) - v_{\hat{T}_\phi}^{\pi_{\mathrm{mix},n}}(s)\pi_{\mathrm{mix},n}(a|s)\Big) r(s,a)\,ds\,da \right| \\
&\le \left| \int \Big(v_T^\pi(s) - v_{\hat{T}_\phi}^{\pi_{\mathrm{mix},n}}(s)\Big) \pi(a|s)\, r(s,a)\,ds\,da \right| \\
&\quad + \left| \int v_{\hat{T}_\phi}^{\pi_{\mathrm{mix},n}}(s)\Big(\pi(a|s) - \pi_{\mathrm{mix},n}(a|s)\Big) r(s,a)\,ds\,da \right| \\
&\le r_{\max} \|v_T^\pi - v_{\hat{T}_\phi}^{\pi_{\mathrm{mix},n}}\|_1 + 2 r_{\max} \, \mathbb{E}_{s \sim v_{\hat{T}_\phi}^{\pi_{\mathrm{mix},n}}}\Big[D_{\mathrm{TV}}\big(\pi(\cdot|s) \,\big\|\, \pi_{\mathrm{mix},n}(\cdot|s)\big)\Big].
\end{aligned}
\tag{32}
$$

A state-visitation bound gives

$$\|v_T^\pi - v_{\hat{T}_\phi}^{\pi_{\mathrm{mix},n}}\|_1 \leq 2\gamma \, \mathbb{E}_{(s,a)\sim\rho_T^\pi}\Big[D_{\mathrm{TV}}\big(T(\cdot|s,a) \,\big\|\, \hat{T}_\phi(\cdot|s,a)\big)\Big]$$
$$+ \gamma \, \mathrm{Vol}(\mathcal{S}) \sum_{i=0}^n w_i^n \, D_{\mathrm{TV}}\Big(\rho_T^\pi(s,a) \,\Big\|\, \rho_T^{\pi_i}(s,a)\Big), \tag{33}$$

and by convexity of $D_{\mathrm{TV}}$ in the second argument and $\pi_{\mathrm{mix},n} = \sum_{i=0}^n w_i^n \pi_i$,

$$\mathbb{E}_{s\sim v_{\hat{T}_\phi}^{\pi_{\mathrm{mix},n}}}\Big[D_{\mathrm{TV}}\big(\pi(\cdot|s) \,\big\|\, \pi_{\mathrm{mix},n}(\cdot|s)\big)\Big] \leq \sum_{i=0}^n w_i^n \, \mathbb{E}_{s\sim v_{\hat{T}_\phi}^{\pi_{\mathrm{mix},n}}}\Big[D_{\mathrm{TV}}\big(\pi(\cdot|s) \,\big\|\, \pi_i(\cdot|s)\big)\Big]. \tag{34}$$

Substituting the definitions of $\xi_{\rho_i}$ and $\xi_{\pi_i}$ and combining the above inequalities yields the desired result. $\qquad\square$

**Lemma B.2** (On-Policy Decomposition of Occupancy Measure TV). *Let* $\rho_{\hat{T}_\phi}^\pi(s,a) = v_{\hat{T}_\phi}^\pi(s)\pi(a \mid s)$ *denote the discounted state–action occupancy measure induced by* $\pi$ *under model* $\hat{T}_\phi$. *Then, for any* $\pi_{\mathrm{D}}$ *and* $\pi$,

$$D_{\mathrm{TV}}\Big(\rho_{\hat{T}_\phi}^{\pi_{\mathrm{D}}}(s,a) \,\|\, \rho_{\hat{T}_\phi}^\pi(s,a)\Big) \leq D_{\mathrm{TV}}\Big(v_{\hat{T}_\phi}^\pi(s) \,\|\, v_{\hat{T}_\phi}^{\pi_{\mathrm{D}}}(s)\Big)$$
$$+ \, \mathbb{E}_{s\sim v_{\hat{T}_\phi}^\pi}\Big[D_{\mathrm{TV}}(\pi(\cdot \mid s) \,\|\, \pi_{\mathrm{D}}(\cdot \mid s))\Big]. \tag{35}$$

*Proof.* Since total variation is symmetric,

$$D_{\mathrm{TV}}\Big(\rho_{\hat{T}_\phi}^{\pi_{\mathrm{D}}} \,\|\, \rho_{\hat{T}_\phi}^\pi\Big) = D_{\mathrm{TV}}\Big(\rho_{\hat{T}_\phi}^\pi \,\|\, \rho_{\hat{T}_\phi}^{\pi_{\mathrm{D}}}\Big). \tag{36}$$

Using the factorization $\rho_{\hat{T}_\phi}^\pi(s,a) = v_{\hat{T}_\phi}^\pi(s)\pi(a \mid s)$ and $\rho_{\hat{T}_\phi}^{\pi_{\mathrm{D}}}(s,a) = v_{\hat{T}_\phi}^{\pi_{\mathrm{D}}}(s)\pi_{\mathrm{D}}(a \mid s)$, we have

$$2D_{\mathrm{TV}}\Big(\rho_{\hat{T}_\phi}^\pi, \rho_{\hat{T}_\phi}^{\pi_{\mathrm{D}}}\Big) = \int \sum_a \Big|v_{\hat{T}_\phi}^\pi(s)\pi(a \mid s) - v_{\hat{T}_\phi}^{\pi_{\mathrm{D}}}(s)\pi_{\mathrm{D}}(a \mid s)\Big| \, ds. \tag{37}$$

Add and subtract $v_{\hat{T}_\phi}^{\pi_{\mathrm{D}}}(s)\pi(a \mid s)$ and apply triangle inequality:

$$\sum_a \Big|v_{\hat{T}_\phi}^\pi(s)\pi(a \mid s) - v_{\hat{T}_\phi}^{\pi_{\mathrm{D}}}(s)\pi_{\mathrm{D}}(a \mid s)\Big|$$
$$\leq \sum_a \Big|\big(v_{\hat{T}_\phi}^\pi(s) - v_{\hat{T}_\phi}^{\pi_{\mathrm{D}}}(s)\big)\pi(a \mid s)\Big| + \sum_a \Big|v_{\hat{T}_\phi}^{\pi_{\mathrm{D}}}(s)\big(\pi(a \mid s) - \pi_{\mathrm{D}}(a \mid s)\big)\Big|. \tag{38}$$

For the first term, $\sum_a \pi(a \mid s) = 1$, hence

$$\sum_a \Big|\big(v_{\hat{T}_\phi}^\pi(s) - v_{\hat{T}_\phi}^{\pi_{\mathrm{D}}}(s)\big)\pi(a \mid s)\Big| = \Big|v_{\hat{T}_\phi}^\pi(s) - v_{\hat{T}_\phi}^{\pi_{\mathrm{D}}}(s)\Big|. \tag{39}$$

For the second term,

$$\sum_a v_{\hat{T}_\phi}^{\pi_{\mathrm{D}}}(s) \, |\pi(a \mid s) - \pi_{\mathrm{D}}(a \mid s)| = 2v_{\hat{T}_\phi}^{\pi_{\mathrm{D}}}(s) \, D_{\mathrm{TV}}(\pi(\cdot \mid s), \pi_{\mathrm{D}}(\cdot \mid s)). \tag{40}$$

Integrating over $s$ and dividing by 2 yields

$$D_{\mathrm{TV}}\Big(\rho_{\hat{T}_\phi}^\pi, \rho_{\hat{T}_\phi}^{\pi_{\mathrm{D}}}\Big) \leq D_{\mathrm{TV}}\Big(v_{\hat{T}_\phi}^\pi, v_{\hat{T}_\phi}^{\pi_{\mathrm{D}}}\Big) + \mathbb{E}_{s\sim v_{\hat{T}_\phi}^{\pi_{\mathrm{D}}}}\Big[D_{\mathrm{TV}}(\pi(\cdot \mid s), \pi_{\mathrm{D}}(\cdot \mid s))\Big]. \tag{41}$$

Finally, swap the roles of $(\pi, \pi_{\mathrm{D}})$ using symmetry of TV to obtain the stated form with expectation under $v_{\hat{T}_\phi}^\pi$. $\qquad\square$

**Lemma B.3** (Bellman Flow). *Let $\mu$ be the initial-state distribution and $\gamma \in (0,1)$. Define the discounted state visitation distribution under $\hat{T}_\phi$ as*

$$v_{\hat{T}_\phi}^\pi \triangleq (1-\gamma) \sum_{t=0}^\infty \gamma^t \mu P_\pi^t. \tag{42}$$

*Then $v_{\hat{T}_\phi}^\pi$ satisfies*

$$v_{\hat{T}_\phi}^\pi = (1-\gamma)\mu + \gamma\, v_{\hat{T}_\phi}^\pi P_\pi. \tag{43}$$

**Lemma B.4** (TV non-expansiveness of the transition operator). *Let $\hat{T}_\phi$ be a fixed dynamics model and $\pi$ be any policy. Define the induced state transition kernel*

$$P_\pi(s' \mid s) \triangleq \int \pi(a \mid s)\, \hat{T}_\phi(s' \mid s, a)\, da. \tag{44}$$

*Then for any probability measures $p, q$ over the state space,*

$$\|pP_\pi - qP_\pi\|_1 \le \|p - q\|_1. \tag{45}$$

*Equivalently, there exists a constant $L_s \in [0,1]$ such that*

$$\|pP_\pi - qP_\pi\|_1 \le L_s \|p - q\|_1, \tag{46}$$

*where $L_s$ denotes the worst-case TV Lipschitz constant of the induced transition operator. For finite-horizon $k$-step rollouts we define*

$$L_s^{(k)}(\hat{T}_\phi) \triangleq \sup_{p \ne q} \frac{\|pP_\pi^k - qP_\pi^k\|_1}{\|p - q\|_1} \in [0,1]. \tag{47}$$

*Empirically, in Dyna-style MBRL $L_s^{(k)}(\hat{T}_\phi)$ is often smaller than 1 for moderate $k$, yielding an effective contraction over finite horizons.*

**Lemma B.5** (Compounding Error Bound on State Distributions). *For any two policies $\pi_1, \pi_2$, the induced state visitation distributions satisfy*

$$D_{\mathrm{TV}}\left(v_{\hat{T}_\phi}^{\pi_1}, v_{\hat{T}_\phi}^{\pi_2}\right) \le \frac{\gamma}{1 - \gamma L_s}\, \mathbb{E}_{s \sim v_{\hat{T}_\phi}^{\pi_2}}\left[D_{\mathrm{TV}}(\pi_1(\cdot \mid s), \pi_2(\cdot \mid s))\right]. \tag{48}$$

*Proof.* Let $v_i = v_{\hat{T}_\phi}^{\pi_i}$ and $P_i = P_{\pi_i}$ for $i \in \{1, 2\}$. From Lemma B.3,

$$v_1 - v_2 = \gamma\Big((v_1 - v_2)P_1 + v_2(P_1 - P_2)\Big). \tag{49}$$

Taking the $L^1$ norm and applying the triangle inequality yields

$$\|v_1 - v_2\|_1 \le \gamma\|(v_1 - v_2)P_1\|_1 + \gamma\|v_2(P_1 - P_2)\|_1. \tag{50}$$

For short-horizon k-rollout, the transition of dynamics model is limited. Therefore, we could use Lemma B.4,

$$\|(v_1 - v_2)P_1\|_1 \le L_s \|v_1 - v_2\|_1. \tag{51}$$

Rearranging gives

$$(1 - \gamma L_s)\|v_1 - v_2\|_1 \le \gamma\|v_2(P_1 - P_2)\|_1. \tag{52}$$

For any state $s$,

$$P_1(\cdot \mid s) - P_2(\cdot \mid s) = \int (\pi_1(a \mid s) - \pi_2(a \mid s))\, \hat{T}_\phi(\cdot \mid s, a)\, da, \tag{53}$$

and since $\|\hat{T}_\phi(\cdot \mid s, a)\|_1 = 1$,

$$\|P_1(\cdot \mid s) - P_2(\cdot \mid s)\|_1 \le \|\pi_1(\cdot \mid s) - \pi_2(\cdot \mid s)\|_1. \tag{54}$$

Therefore,

$$\|v_2(P_1 - P_2)\|_1 \le \mathbb{E}_{s \sim v_2}\big[\|\pi_1(\cdot \mid s) - \pi_2(\cdot \mid s)\|_1\big]. \tag{55}$$

Using $\|p - q\|_1 = 2D_{\mathrm{TV}}(p, q)$ completes the proof. $\qquad\square$

# C. Consistency Coefficient of the Dynamics Model

## Rollout-Relevant Consistency Coefficient

Let $\hat{T}_\phi(\cdot \mid s, a)$ denote the learned dynamics model and $\pi_\phi(a \mid s)$ the current policy. For a rollout horizon $k$, we consider the set of states that are reachable within $k$ model steps under $(\pi_\phi, \hat{T}_\phi)$, starting from $s_0 \sim \mathcal{D}_{\mathrm{env}}$. We denote this rollout-relevant state region by $\mathcal{S}_k$.

Intuitively, $\mathcal{S}_k$ captures the subset of the state space that is actually encountered during short-horizon model rollouts and therefore governs the practical impact of model errors on policy learning.

We define the *rollout-relevant consistency coefficient* $L_s^{(k)}$ as the maximum total-variation distance between $k$-step state distributions induced by the learned model and policy, over all pairs of initial states in $\mathcal{S}_k$. Formally,

$$L_s^{(k)} \triangleq \sup_{s_1, s_2 \in \mathcal{S}_k} D_{\mathrm{TV}}\left( P_{\pi_\phi, \hat{T}_\phi}^{(k)}(\cdot \mid s_1), P_{\pi_\phi, \hat{T}_\phi}^{(k)}(\cdot \mid s_2) \right), \tag{56}$$

where $P_{\pi_\phi, \hat{T}_\phi}^{(k)}(\cdot \mid s)$ denotes the state distribution obtained after rolling out the learned model for $k$ steps starting from $s$ under policy $\pi_\phi$. A small value of $L_s^{(k)}$ implies that nearby initial states induce similar rollout distributions, whereas a large value indicates strong amplification of local perturbations across rollout steps.

## Useful Model Error and a Consistency Perspective

A recurring empirical observation in model-based reinforcement learning is that models with inferior one-step prediction accuracy may nevertheless yield superior policy performance. This indicates that the magnitude of one-step prediction error alone is insufficient to characterize the impact of model inaccuracies on policy optimization. One plausible explanation is that certain model errors are useful: although the learned dynamics may deviate from the true environment at the transition level, such deviations can smooth the effective dynamics along policy rollouts, improving robustness to state perturbations and enhancing generalization (Young et al., 2022).

This perspective aligns naturally with the role of the sensitivity coefficient $L_s^{(k)}$ in our analysis. Smoother rewards or transition dynamics reduce rollout sensitivity, thereby narrowing the gap between model-based rollouts and real environment trajectories. In relatively simple continuous-control domains (e.g., Hopper), the dominant source of performance discrepancy often arises from rollout sensitivity rather than raw one-step prediction error. Motivated by this observation, we introduce a *consistency regularization* mechanism that explicitly targets the geometric smoothness of short-horizon model rollouts.

## Consistency-Regularized Dynamics Training

Given a real state $s_t$ sampled from the environment replay buffer, we generate a short model rollout autoregressively. The rollout is initialized at $\tilde{s}_t = s_t$. At each step, an action is sampled from the current policy $\tilde{a}_{t+j} \sim \pi_\phi(\cdot \mid \tilde{s}_{t+j})$, and the next state is sampled from the learned dynamics model $\tilde{s}_{t+j+1} \sim \hat{T}_\phi(\cdot \mid \tilde{s}_{t+j}, \tilde{a}_{t+j})$.

To quantify the local geometry of short trajectories, we adopt a second-order finite-difference approximation. For a real trajectory segment $(s_t, s_{t+1}, s_{t+2})$, the discrete curvature at time $t+1$ is defined as

$$\kappa_{t+1} = s_{t+2} - 2s_{t+1} + s_t. \tag{57}$$

Similarly, for a model rollout $(\tilde{s}_t, \tilde{s}_{t+1}, \tilde{s}_{t+2})$, the corresponding curvature is

$$\tilde{\kappa}_{t+1} = \tilde{s}_{t+2} - 2\tilde{s}_{t+1} + \tilde{s}_t, \qquad \tilde{s}_t = s_t. \tag{58}$$

Unlike one-step prediction losses, curvature directly constrains the geometric shape of short-horizon rollouts rather than individual transitions, making it particularly well-suited for controlling rollout sensitivity.

The dynamics model is trained using a likelihood-based objective that matches multi-step transitions observed in the environment replay buffer. Let $m$ denote the maximum rollout horizon used for model fitting. The multi-step negative log-likelihood loss is given by

$$\mathcal{L}_{\mathrm{dyn}} = -\frac{1}{m} \sum_{k=1}^{m} \mathbb{E}_{\mathcal{D}_{\mathrm{env}}}\left[ \log \hat{T}_\phi\big(s_{t+k}, r_{t+k} \mid s_t, a_{t:t+k-1}\big) \right]. \tag{59}$$

When rewards are modeled separately, this objective can be restricted to state transitions without loss of generality.

To align regularization with the policy-dependent analysis used throughout this work, we define a curvature consistency term under the model-induced occupancy measure $\rho^\pi_{\hat{T}_\phi}$. Specifically,

$$\mathcal{R}_\kappa(\pi, \hat{T}_\phi) = \mathbb{E}_{(s_t, a_t) \sim \rho^\pi_{\hat{T}_\phi}} \left[ \|\tilde{\kappa}_{t+1} - \kappa_{t+1}\|^2_M \right], \tag{60}$$

which penalizes deviations between the curvature of model-generated rollouts and that of real environment trajectories under the same policy-induced distribution. The final training objective is

$$\mathcal{L}_{\text{cons}} = \mathcal{L}_{\text{dyn}} + \lambda_\kappa \mathcal{R}_\kappa(\pi, \hat{T}_\phi), \tag{61}$$

where $\lambda_\kappa \geq 0$ controls the strength of curvature regularization. In all experiments, we set $M = I$, corresponding to uniform penalization across state dimensions.

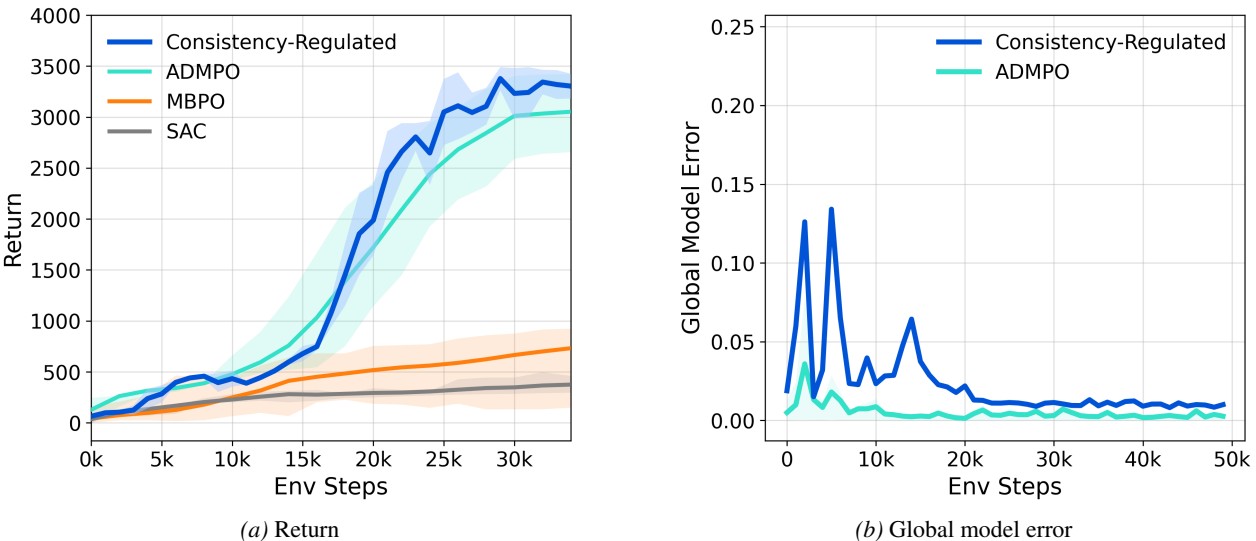

*(a)* Return        *(b)* Global model error

*Figure 8.* Comparison of return curves on Hopper task.

**Hopper.** We evaluate consistency regularization on the Hopper environment by augmenting the dynamics model training objective with $\mathcal{L}_{\text{cons}}$. Figure 8 shows the results. Although the resulting model exhibits worse one-step prediction accuracy than ADMPO, policy learning converges significantly faster, reaching near-optimal performance (approximately 3500 return) within 30k environment steps. This behavior is expected: enforcing local smoothness implicitly alters the fine-grained structure of the environment dynamics and degrades exact next-state prediction. However, by reducing the rollout sensitivity coefficient $L_s^{(k)}$, policy improvements realized under the learned model are more consistently reflected in the real environment, thereby accelerating learning.

**Failure in Complex Environments.** The smoothness assumption underlying consistency regularization is inherently regime-dependent. In higher-dimensional environments such as Humanoid, enforcing curvature constraints disrupts the intrinsic structure of the dynamics, dramatically increases one-step prediction error, and prevents stable convergence. In these settings, compounding errors grow rapidly, ultimately degrading rather than improving environment consistency.

These results indicate that once model prediction error is sufficiently small, policy performance is primarily governed by rollout sensitivity and the amplification of compounding errors, as captured by $L_s^{(k)}$.

For more complex environments, effective policy optimization requires dynamics models that simultaneously achieve accurate one-step prediction and strong environment-level consistency. One promising direction is to map high-sensitivity environments into latent spaces with reduced transition sensitivity, as explored in prior work such as Dreamer. This perspective can be interpreted as optimizing environment-level consistency, but lies beyond the scope of the present work.

## D. Validating Prediction Error as a Proxy for Policy Mismatch

This appendix validates the key intuition that dynamics model prediction error can serve as a practical proxy for policy-induced mismatch. Figure 9 visualizes model prediction error against policy shift on Hopper.

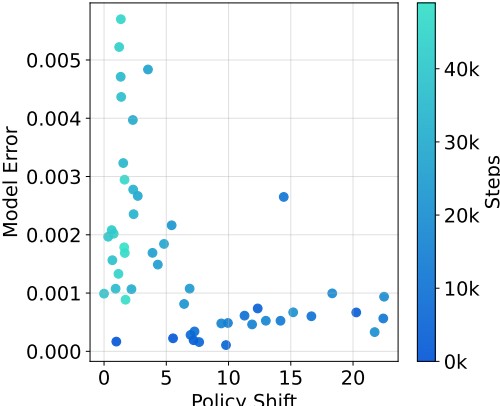

*Figure 9.* Model error versus policy shift on Hopper. The 50k-step dynamics model is used to evaluate prediction error, and policy shift is measured as the KL divergence between the 50k-step policy and historical policies.

We use the 50k-step dynamics model to evaluate prediction error, and measure policy shift as the KL divergence between the 50k-step policy and historical policies. Prediction error is not identical to policy shift. However, high-error samples are concentrated in regions closer to the current policy, whereas samples from larger-shift historical policies often have lower error because they have been better covered during model training. This supports our view that prediction error highlights underfitted, policy-relevant regions without explicit policy-distance estimation.

The above result provides a direct diagnostic of why prediction error can be informative. We next analyze how this signal affects the replay behavior of PMER during dynamics model training.

**Experimental setup.** During training, we periodically (every 1k environment steps) record the policy checkpoints and the corresponding dynamics model checkpoints. At iteration $n > 1$, we consider the historical policy sequence $\{\pi_1, \ldots, \pi_n\}$. Each policy $\pi_i$ is used to independently collect a set of transitions $\mathcal{T}_i$. Using the dynamics model $\hat{T}_\phi^n$ at iteration $n$, we evaluate the prediction error on each transition set. The prediction error for each transition is computed consistently with Eq. 8. For each policy $\pi_i$, we compute the average L2 prediction error

$$\hat{\psi}_i = \frac{1}{K} \sum_{k=1}^{K} \psi_k, \tag{62}$$

where $K = 1000$ denotes the number of sampled transitions.

**Error-weighted policy distance.** To quantify policy-induced distribution shift, we measure the distance between the current policy $\pi_n$ and each historical policy $\pi_i$ using the KL divergence, defined as

$$\xi_{\pi_i} = \frac{1}{K} \sum_{k=1}^{K} D_{\mathrm{KL}}\left(\pi_n(\cdot \mid s_k) \,\|\, \pi_i(\cdot \mid s_k)\right), \tag{63}$$

where $\{s_k\}_{k=1}^{K}$ are states sampled from the replay buffer.

We construct an error-based weight vector by linearly normalizing the prediction errors:

$$w_i = \frac{\hat{\psi}_i}{\sum_{j=1}^{n} \hat{\psi}_j}. \tag{64}$$

Using these weights, the error-weighted average policy distance is computed as

$$d_{\text{error}} = \sum_{i=1}^{n} w_i \, \xi_{\pi_i} = w^\top \boldsymbol{\xi}, \tag{65}$$

where $\boldsymbol{\xi} = [\xi_{\pi_1}, \ldots, \xi_{\pi_n}]$. For comparison, we also compute the uniformly weighted distance $d_{\text{uniform}}$ by setting $w_i = 1/n$.

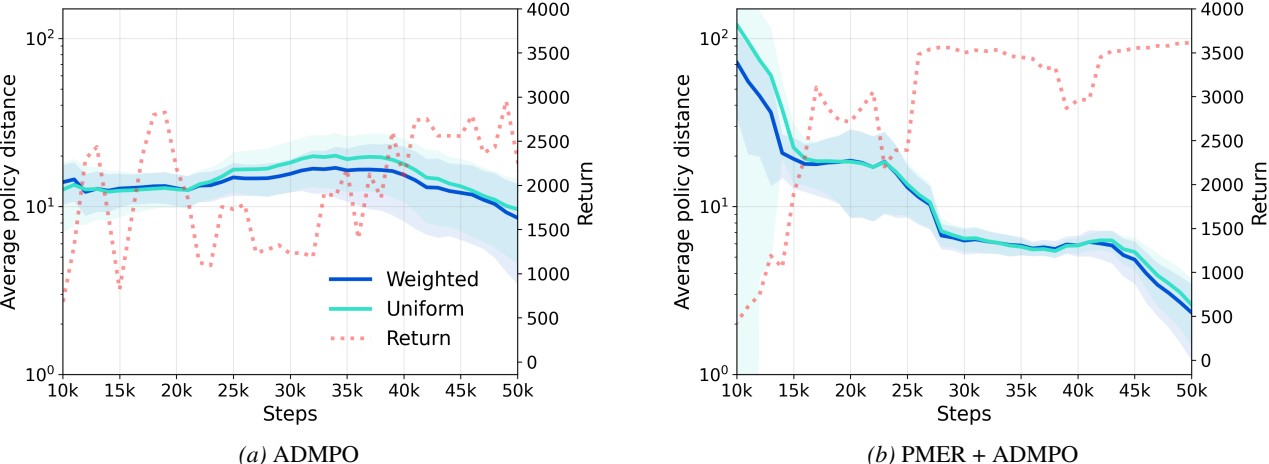

*(a)* ADMPO          *(b)* PMER + ADMPO

*Figure 10.* Error-weighted and uniformly weighted average policy distance measured during training on Hopper task.

**Results and discussion.** The results are shown in Figure 10. During the early stage of training, the error-weighted distance closely matches the uniformly weighted distance. This behavior is expected, as rapid policy updates dominate global policy variation in the early stage, making most historical policies similarly mismatched.

As training progresses, the error-weighted policy distance becomes consistently lower than the uniformly weighted distance. Since prediction error is the only signal used to construct the weights, this result indicates that transitions with larger dynamics model prediction error are more strongly associated with policy-induced distribution shift.

We further repeat the error-weighted policy distance analysis under PMER-enhanced policy training. Unlike the ADMPO baseline, where error-weighted and uniformly weighted policy distances exhibit a clear separation during the middle and late training stages, the two curves largely overlap under PMER-enhanced training. This behavior indicates that policy-induced distribution shift is substantially reduced throughout training when PMER is applied.

We emphasize that this result does not contradict the proxy validation presented in the ADMPO setting. Rather, it reflects the fact that PMER continuously corrects policy-relevant dynamics model errors during training, thereby reducing the discrepancy between historical policies and the current policy. As a consequence, both error-weighted and uniformly weighted policy distances converge to similarly low values.

Together with the ADMPO results, this comparison highlights a self-consistent mechanism: prediction error serves as an effective proxy for identifying policy mismatch, while PMER actively mitigates such mismatch, eventually rendering explicit reweighting unnecessary.

# E. Ablation Study

We provide ablation studies on the prioritization strength $\beta$ and the temporal decay factor $\lambda$ in Figures 11a, 11b, and 11c. For $\beta$, we evaluate different values and find that PMER works well over a broad range, although not all settings are equally effective. The best convergence is observed around $\beta \in [2, 3]$ on both Hopper and Humanoid. When $\beta$ is small, sampling is closer to uniform over the prioritized buffer, so high-error transitions are not emphasized enough, limiting the correction of local mismatch. When $\beta$ is large, sampling concentrates more heavily on the highest-error transitions. While this can better target mismatch-relevant samples, it also reduces sample diversity, which may slow convergence and hurt stability.

For $\lambda$, removing decay leads to clear underperformance because stale high-error samples persist in the buffer, indicating that dynamic decay is necessary. By contrast, overly strong decay suppresses high-error samples too aggressively, making PMER less effective at correcting local mismatch. These results show that moderate prioritization strength and temporal decay are important for balancing mismatch correction and sample diversity.

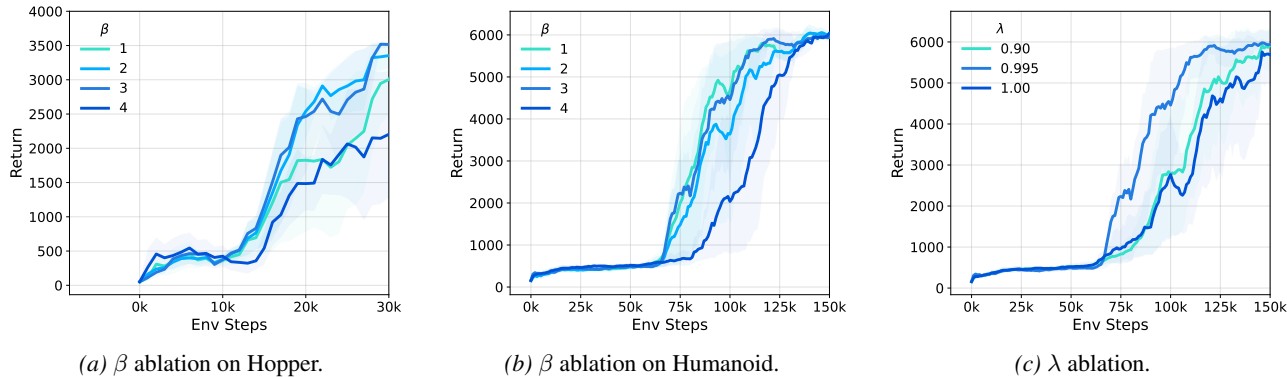

(a) $\beta$ ablation on Hopper.     (b) $\beta$ ablation on Humanoid.     (c) $\lambda$ ablation.

*Figure 11.* Ablation studies on the prioritization strength $\beta$ and temporal decay factor $\lambda$. Moderate $\beta$ values balance mismatch correction and sample diversity, while temporal decay prevents stale high-error samples from dominating model training.

# F. Backbone Selection and Generality

We adopt ADMPO as the backbone algorithm for integrating PMER for three primary reasons. First, ADMPO preserves the core learning paradigm of MBPO, including short-horizon $k$-step model rollouts and SAC-based policy optimization, while introducing only minimal architectural modifications. This allows PMER to be evaluated within a familiar and well-established model-based reinforcement learning framework, ensuring that observed performance differences can be attributed to the replay mechanism rather than fundamental changes in the optimization procedure.

Second, ADMPO replaces the ensemble dynamics model commonly used in MBPO with a single adaptive dynamics model, which substantially reduces compounding errors while maintaining training stability. This design choice is particularly aligned with the motivation of PMER, as our method explicitly targets local model prediction errors and their interaction with policy-induced distribution shift. Using a single dynamics model also enables a clearer analysis of how prioritized replay affects model learning behavior, without the confounding effects introduced by ensemble disagreement.

Finally, adopting ADMPO as the backbone facilitates both fair horizontal comparisons with other MBPO-style algorithms and a focused vertical investigation into the role of dynamics model learning in model-based reinforcement learning. By avoiding additional sources of uncertainty modeling or complex rollout adaptation strategies, ADMPO provides a controlled experimental setting in which the effectiveness of PMER can be isolated and evaluated.

**From MBPO to ADMPO.** PMER was initially developed and validated on the classical MBPO framework, where it already yields substantial gains in early learning speed.

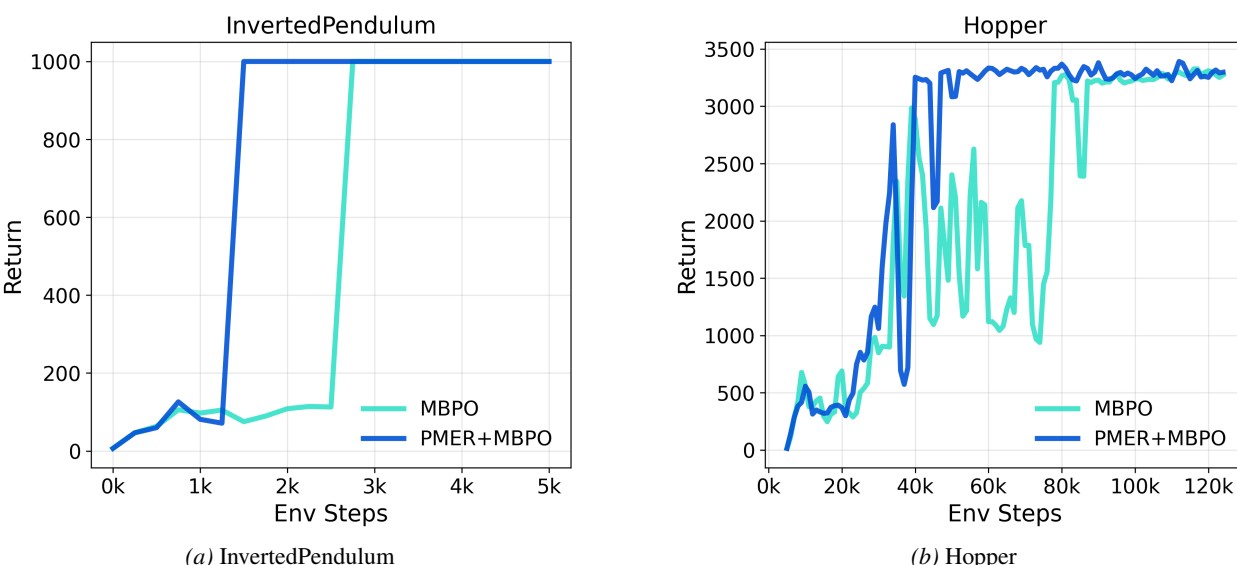

*(a)* InvertedPendulum        *(b)* Hopper

*Figure 12.* Comparison of MBPO and MBPO+PMER on two environments. PMER substantially accelerates early learning on both tasks, reaching near-optimal performance with approximately $2\times$ fewer environment interactions, while preserving comparable asymptotic returns. These results illustrate that the core benefit of PMER is not specific to ADMPO and already emerges in the classical MBPO framework.

Figure 12 shows a representative sanity check on InvertedPendulum and Hopper, in which MBPO+PMER reaches the maximum return with approximately $2\times$ fewer environment interactions than the vanilla MBPO baseline, while preserving the same asymptotic performance.

PMER also reduces the pronounced performance oscillations exhibited by MBPO in the mid-training phase, which are symptomatic of rollout bias and model exploitation. This behavior supports our hypothesis that PMER stabilizes dynamics model learning by repeatedly revisiting high-error transitions, thereby mitigating policy-induced distribution shift and its amplification through the learned model.

Importantly, these gains are achieved without modifying the policy optimization procedure or the rollout strategy, indicating that the primary effect of PMER is to improve the efficiency and stability of dynamics model learning rather than introducing additional sources of exploration or regularization.

We subsequently integrate PMER into ADMPO to study its effect under a more stable and lower-variance dynamics modeling setup. Using ADMPO as the primary backbone allows us to disentangle the effect of prioritized model replay from ensemble uncertainty and rollout adaptation heuristics, and to conduct a more controlled investigation of how error-aware data replay shapes dynamics model learning and policy optimization behavior.

Taken together, the MBPO sanity checks and the ADMPO main experiments suggest a coherent mechanism underlying the empirical gains of PMER. Across both backbones, PMER consistently accelerates early learning and reduces mid-training performance oscillations, without degrading asymptotic returns. This interpretation holds across both ensemble-based MBPO and single-model ADMPO, indicating that the core benefit of PMER is backbone-agnostic and arises from improved dynamics model learning rather than architectural idiosyncrasies or rollout heuristics.

**PMER for MBPO Settings.** When integrating PMER into the classical MBPO framework for the sanity-check experiments on InvertedPendulum and Hopper, we adopt a minimal and fully non-intrusive modification of the original MBPO training pipeline. Specifically, the prioritization weight for each $k$-step transition sequence collected by policy $\pi_n$ is defined as the single-step model prediction error

$$\psi_i^n = \left\| \mu_\theta \left( \mathbf{s}_t^{(i)}, \mathbf{a}_{t:t+k-1}^{(i)} \right) - \begin{bmatrix} \mathbf{s}_{t+k}^{(i)} - \mathbf{s}_t^{(i)} \\ r_{t+k-1}^{(i)} \end{bmatrix} \right\|_2^2, \tag{66}$$

where $k = 1$ in all MBPO experiments, $\mu_\theta(\cdot)$ denotes the mean prediction of the ensemble dynamics model, and the target consists of the observed state increment and reward.

For the ensemble dynamics model, each training batch has shape $[\text{Ensemble}, \text{Batch}, \text{Obs}]$. We compute the squared prediction loss for each ensemble member and then average the losses across the ensemble dimension, yielding a per-sample prioritization weight of shape $[\text{Batch}, \text{Obs}]$. This averaged error is used as the priority score for inserting transitions into the prioritized model buffer.

Importantly, all other components of MBPO remain unchanged, including the policy optimization procedure, rollout horizon, model architecture, and training schedule. PMER only augments the data sampling mechanism for dynamics model learning, without introducing additional regularization, exploration bonuses, or rollout adaptation heuristics.

The hyper-parameters of PMER used in the MBPO sanity-check experiments are summarized in Table 1.

*Table 1.* PMER hyper-parameters for MBPO sanity-check experiments.

| Environment | Prior ratio $\rho$ | Prior buffer size $K$ | Decay rate $\lambda$ | Convergence step |
|---|---|---|---|---|
| InvertedPendulum | 0.05 | 2000 | 0.995 | 2k |
| Hopper | 0.30 | 2000 | 0.995 | 50k |

# G. Hyperparameter Settings

Our hyperparameter settings are provided in Table 2 and Table 3.

*Table 2.* Environment-specific hyperparameter settings. The notation $x \to y$ over $a \to b$ denotes a thresholded linear increasing schedule; that is, the model rollout horizon at environment step $t$ is calculated as $f(t) = \min\left(\max\left(x + \frac{t-a}{b-a}(y-x), x\right), y\right)$.

| Parameter | HalfCheetah | Hopper | Walker2d | Ant | Humanoid |
|---|---|---|---|---|---|
| Training Steps | 200k | 50k | 200k | 200k | 200k |
| Update-to-Data Ratio | | | 20 | | |
| Prior Ratio $\rho$ | | | 0.1 | | |
| Prior Buffer Size | | | 2000 | | |
| Model Rollout Horizon $k$ Schedule | $\frac{1}{0 \to 200k}$ | $\frac{1 \to 15}{0 \to 50k}$ | $\frac{1 \to 10}{0 \to 100k}$ | $\frac{1 \to 5}{10k \to 100k}$ | $\frac{1 \to 10}{10k \to 100k}$ |
| Target Entropy | $-1$ | $-1$ | $-3$ | $-4$ | $-8$ |
| Weight Decay Schedule | | | 0.995 | | |
| $\beta$-Sampling Schedule | | | 3 | | |
| Computational Overhead | ↑ 8% | ↑ 10% | ↑ 12% | ↑ 8% | ↑ 9% |
| Priority Prediction Target $k_p$ | $\{1, 2\}$ | $\{1, 2, 3, 4, 5\}$ | $\{1, 2\}$ | $\{1, 2\}$ | $\{1, 2\}$ |

*Table 3.* Key Online Training Parameters.

| Parameter | Value |
|---|---|
| Environment | MuJoCo |
| Device | NVIDIA RTX 3090 |
| Actor/Critic Learning Rate | $3 \times 10^{-4}$ / $3 \times 10^{-4}$ |
| Entropy Coefficient $\alpha$ | Auto (lr $= 3 \times 10^{-4}$) |
| Actor Update Frequency | 20 |
| Discount Factor $\gamma$ | 0.99 |
| Number of Critics | 2 |
| ARM Hidden Dimension | 200 (Humanoid 400) |
| Batch Size | 256 |
| Warmup Rollouts | 5,000 |
| Replay Buffer Size | $1 \times 10^6$ |
| Real Data Ratio | 0.05 |
| Optimizer | Adam |
| Activation | ReLU |

