# OpenReview forum: "Prioritized Model Experience Replay"
_ICML.cc/2026/Conference — ICML 2026 regular_

### Official Review · Reviewer_ozf7 · 2026-03-13

**Soundness:** 3
**Presentation:** 3
**Significance:** 2
**Originality:** 2
**Overall Recommendation:** 4
**Confidence:** 4

**Summary:**

This paper tackles the problem of distribution mismatch in online model-based reinforcement learning. Indeed, the distribution mismatch issue emanates from the fact that the model is trained on data generated by a mixture of previous policies (the replay buffer), while being used with a current policy that might exhibit a distributional mismatch with the respect to the training data mixture. Previous work have shown that weighting the training data using an estimate of the respective policy mismatch is useful in mitigating this issue. This paper suggest a more lightweight approach where the mixture weights are estimated directly from the model global prediction error, avoiding additional computational cost to estimate the policy mismatch terms. A theoretical analysis is conducted to prove a return bound that depend on the policy mixture weights. Finally, the resulting algorithm is shown to achieve similar or better results in terms of the return in continuous control RL tasks.

**Compliance With Llm Reviewing Policy:**

Affirmed.

**Final Justification:**

I would like to thank the authors for their rebuttal. Most of my concerns are resolved through the new set of experiments, therefore I'm raising the score to weak accept. I still believe that the paper would benefit from a major revision by revisiting the current narrative based on the experiments added during rebuttal (which have to cover more environments, etc). Finally, about the random weights baseline, if I understood correctly the authors last response, this baseline is computed by randomly sampling transitions, which is no different than the equal-weight baseline. I suggest that the authors reconsider how they compute the random-weight baseline if they include these experiments. Otherwise, the weight-profile comparison between PMER and PDML is sufficient to answer the question about how relevant are the actual weights.

**Key Questions For Authors:**

* Did the authors explore controlling the second term of the bound as well, for example through an explicit Lipschitz constant regularization of the learned dynamics model?
* Given the formulation of the mixture weights dependent term $\mathcal{M}(\omega^n)$ in the appendix, do the weights as defined in PDML $\omega_i^n = \frac{\xi^{-1}(\pi_i)}{\sum_{m=1}^{n} \xi^{-1}(\pi_m)}$ correspond to the theoretically optimal weights that minimize $\mathcal{M}(\omega^n)$?
* Instead of the error profile shown in Figure 9, maybe it could be interesting to visualize the correlation between respective model errors and policy mismatch terms, in order to strengthen the authors' current intuition about model prediction error.

**Limitations:**

yes

**Strengths And Weaknesses:**

**Strengths:**
* The paper provides a simplified version of a policy mismatch-based approach that relies on computationally expensive estimation of the policy mismatch terms in order to compute the mixture weights. This simplification improves the current algorithm in terms of ease of implementation and computational overhead, all while retaining similar or better performance as measured by the return of the underlying agents.

**Weaknesses:**
* The weights that are obtained through the respective model prediction errors of previous policies are not compared to the policy distribution-based weights (as in PDML). Indeed, because the model prediction error is used as proxy for the policy mismatch in estimating the mixture effect, an interesting question to ask is whether the two approaches result in practically the same weights. This is especially relevant because the return analysis show very close results between PMER and PDML in almost all environments. Another way to study this issue is to investigate sampling random weights (that are close the actual weight profiles to some extent), and observe the impact on the final return. This would help clarify whether the model prediction error-based weights are actually useful or just within noise levels of the return achieved by the baseline.
* The empirical results on which the mixture-policy effect is substituted with model prediction error-based weights are not significant nor conclusive. Indeed, Figure 9 in Appendix D show almost overlapping error profiles with no significant difference between model error induced weights and uniform weights.
* Overall, the approach doesn't show substantial improvements over the closest relatives (ADMPO and especially PDML). This is not a reason for rejection, however, given the simplicity of the approach I think that the paper would highly benefit from more experiments / environments to further prove the effectiveness of the approach.

---

> ### Author Rebuttal · Authors · 2026-03-31
>
> Thank you for this thoughtful feedback! Our response is below, and the linked file is available [here](https://anonymous.4open.science/api/repo/PMER/file/1.pdf).
>
> > **Q1: Random weight assignment experiment.**
>
> This is a helpful suggestion, and we followed it with an additional experiment on the Humanoid task. We compare three weighting schemes:
> (1) uniform weights: no PMER.
> (2) random-weight proxy: we randomly select transitions for the Prioritized Model Experience Buffer instead of using model prediction error.
> (3) PMER: the proposed method with the same setting as in the paper.
>
> Figure 8 of the linked file shows that random weighting yields little gain over uniform replay, whereas PMER gives a clear improvement.
>
> > **Q2: Figure 9 shows overlapping error profiles.**
>
> We may not have explained the setup clearly enough. Under ADMPO, the error-weighted and uniformly weighted distances separate in the middle and late stages of training, showing that prediction error highlights regions different from those emphasized by uniform replay. Under PMER, the two profiles become much closer and decrease together.
> This is expected: the separation in ADMPO supports prediction error as a useful proxy, while the later overlap under PMER indicates that the mismatch has already been reduced, as discussed in Appendix D.
>
> Figure 9 also uses a logarithmic scale, which visually compresses the gap. Under a linear scale, the ADMPO separation would appear more pronounced. Overall, we interpret the figure as showing that the curves remain more separated without PMER, and become more aligned once PMER is applied.
>
> > **Q3: PMER's originality.**
>
> We believe PMER is not simply a substitute for PDML: PDML explicitly reweights historical data via policy-distance estimation, whereas PMER uses prediction error as an implicit proxy in dynamics training. Moreover, by using a dynamically updated buffer instead of reweighting the full replay set, PMER reduces overhead and is mechanistically distinct from prior replay-based designs.
>
> PMER emphasizes harder, policy-relevant samples, rather than focusing only on regions induced by the most recent policy updates. This mitigates oscillatory training from excessive local optimization and improves rollout stability. These gains are supported by improved stability, reduced mismatch, and over 20% average data efficiency improvement across environments (Figures 5–7; Appendix E).
>
> > **Q4: More experiments / environments to further demonstrate the effectiveness.**
>
> Our main experiments follow standard MBRL evaluation protocols and use the same MuJoCo benchmarks as prior work for fair comparison. We agree that more challenging domains would further strengthen the paper; accordingly, we additionally evaluate PMER in a Dreamer-v3 framework, where it shows gains on harder tasks (Figure 9, linked file). Please also see our response to Reviewer Phft.
>
> > **Q5: Did you explore explicit regularization of the second term?**
>
> We agree that explicit regularization of the second term is an interesting direction. However, we believe this is beyond the main scope of this paper. The sensitivity term is largely determined by model design, and PMER is intentionally proposed as a lightweight data-level method that does not modify the model architecture or training objective. Similarly, explicit regularization of the consistency term is not part of the current method, which is also why we group these factors into the model capability term.
>
> That said, in Appendix C we explore a consistency-regularized loss that constrains short-horizon rollout curvature and thus indirectly controls the effective consistency. It helps on Hopper but is unreliable on more complex environments, so we view it as a complementary extension rather than a core part of PMER.
>
> > **Q6: Do the weights as defined in PDML minimize $\\mathcal{M}(w^n)$?**
>
> Not exactly. This corresponds to Proposition 3.2 in PDML, which shows that, compared with uniform weighting, assigning weights negatively related to the state-action visitation shift $\\xi_{\\rho_i}$ and policy shift $\\xi_{\\pi_i}$ can reduce the performance gap. The corresponding argument is provided in Appendix E of PDML.
>
> > **Q7: Can you visualize the correlation between model error and policy shift?**
>
> To help clarify this point, we conducted an additional experiment shown in Figure 1 of the linked file. Higher prediction error tends to appear in regions closer to the current policy, while samples associated with larger policy shift, i.e., older historical policies, exhibit lower error due to repeated training and better coverage. Thus, prioritizing high-error transitions implicitly emphasizes samples closer to the current policy distribution.
>
> ---
> Overall, we believe PMER provides a simple and scalable way to improve data efficiency in future MBRL research. We appreciate your careful review and useful feedback, and hope that the additional experiments and clarifications help address your concerns.

---

> > ### Author Rebuttal · Reviewer_ozf7 · 2026-04-03
> >
> > I would like to thank the reviewers for their efforts. My concerns are partially resolved, therefore I'm maintaining the original score. Regarding Q1, the question is more about comparing the weight profile induced by model errors and the weight profile induced by policy shift that is used in PDML. The random weights was an alternative suggestion, which has to show average +- confidence interval of the return achieved by multiple randomly selected weight profiles in order to provide insights about the actual performance of your method compared to random weights. Regarding Q2, I agree that the decreasing policy distance correlates well with the increase observed in the return. However the separation of uniform and optimal weight profiles is not statistically significant for the baseline and therefore the separation argument can not be made to interpret these results in my opinion.

---

> > > ### Author Response · Authors · 2026-04-04
> > >
> > > Thank you for your reply and follow-up questions. We address the remaining concerns below. Please let us know if you have any further comments or concerns. The follow-up linked file is provided [here](https://anonymous.4open.science/api/repo/PMER-2/file/2.pdf).
> > >
> > > > **Q8: Regarding the random-weight baseline profiles in Rebuttal.**
> > >
> > > We agree that the random-weight comparison should be reported over multiple independently sampled random profiles with summary statistics.
> > >
> > > To address this, we conducted multiple random-weight runs and report the raw training return curves of all seeds in Figure 10 of the [linked file](https://anonymous.4open.science/api/repo/PMER-2/file/2.pdf).
> > > Since the main emphasis of our paper is convergence speed and data efficiency rather than asymptotic return, we use the number of training steps required to reach a fixed convergence threshold as the main summary statistic. The random-weight baseline used in the rebuttal phase requires 144.7k $\\pm$ 27.4k steps (95% CI, 3 seeds) to reach convergence, whereas PMER requires 106.0k $\\pm$ 9.0k steps, corresponding to about a 26.7% reduction in convergence steps on Humanoid.
> > >
> > > In our setup, the random-weight proxy is implemented by randomly selecting transitions into the Prioritized Model Experience Buffer. As a result, different random seeds lead to different sampled transition subsets and thus different effective random weight profiles.
> > > Repeating this baseline across seeds provides a comparison over multiple independently sampled random profiles.
> > >
> > > > **Q9: Direct weight profile comparison.**
> > >
> > > We understand that your main concern is whether PMER induces a weighting pattern over historical policies that is comparable to the one defined by policy shift in PDML. To address this point more directly, we added an additional comparison experiment in Figure 11 of the linked file.
> > >
> > > Specifically, PDML defines historical-policy weights explicitly as an inverse function of policy shift, followed by normalization.
> > > For PMER, we construct an effective weight proxy by using the saved policy checkpoints to collect transitions, store them in the replay buffer, train the dynamics model for one round, and record how often transitions from each historical policy are replayed for model learning. After normalizing these cumulative frequencies, we obtain PMER’s relative sampling preference over historical policies.
> > >
> > > Although the two quantities are not identical by definition, Figure 11 shows broadly similar trends, especially in the middle and late stages of training. The early-stage profile is noisier, which is expected before the prioritized buffer stabilizes.
> > > However, PMER is not intended to exactly reproduce PDML’s explicit weighting rule. As emphasized in Q3, it prioritizes harder, policy-relevant samples rather than only regions induced by the most recent policy updates, which naturally leads to a different allocation profile.
> > >
> > > > **Q10: Clarification on the interpretation of Q2.**
> > >
> > > We fully understand this concern.
> > > We agree that the separation between the uniform and optimal weight profiles in Q2, by itself, is not sufficient to support a strong statistical interpretation. Our intention there was to provide a qualitative diagnostic of how the weighting effect evolves during training, and we will revise the wording accordingly.
> > >
> > > To address your concern more directly, we added the new experiment in Q7 (Figure 1 in the linked file), which explicitly visualizes the relationship between model prediction error and policy shift.
> > > We view this new result as more direct evidence for our key intuition: prediction error is not identical to policy shift, but it serves as a practical proxy by highlighting underfitted regions that are more relevant to the current policy.
> > >
> > > > **Q11: Originality.**
> > >
> > > We believe the paper’s originality lies in both its perspective and its method. It identifies dynamics model learning mismatch as a key source of instability in Dyna-style MBRL and provides new insights into why error-based replay is effective: prediction error acts as a practical proxy for policy-induced mismatch, and replaying high-error transitions helps the dynamics model focus on underfitted, policy-relevant regions. PMER operationalizes this insight through a lightweight replay mechanism.
> > >
> > > ---
> > > Overall, we have added:
> > > - random-weight experiments with mean $\\pm$ 95% confidence intervals across seeds, showing that PMER’s data-efficiency gains are not explained by noise alone;
> > > - a visualization of the correlation between model prediction error and policy shift, supporting the proxy intuition;
> > > - a policy-weight comparison experiment between PDML and PMER, showing a similar emphasis pattern over historical policies.
> > >
> > > We hope these additional analyses resolve the remaining concerns and make the role of prediction-error-based prioritization clearer. We would appreciate it if you could take these new results into account in your final assessment.

---

### Official Review · Reviewer_Phft · 2026-03-13

**Soundness:** 3
**Presentation:** 4
**Significance:** 3
**Originality:** 3
**Overall Recommendation:** 5
**Confidence:** 4

**Summary:**

In recent years, model-based reinforcement learning algorithms have shown promising sample efficiency across many RL benchmarks.  These MBRL algorithms are prone to challenges such as poor exploration, objective mismatch, and model learning mismatch. This paper addresses the model learning mismatch by proposing a method to prioritize the model errors. The PMER approach enjoys a plug-n-play modification over standard MBRL algorithms. The theoretical results provide a finite time bound on the performance gap of the policy on the learned model and the true model. The empirical results demonstrate improved stability and, consequently, better sample complexity.

**Compliance With Llm Reviewing Policy:**

Affirmed.

**Final Justification:**

The paper presents an elegant solution for improving the performance of MBRL algorithms. I appreciate that the authors provided additional results using the DreamerV3 architecture during the rebuttal phase. I recommend that the authors incorporate these results, along with more extensive examples, into the final manuscript. I have updated my score to Accept.

**Key Questions For Authors:**

- Can author provide a justification of using ADMPO as a baseline and building on it. Perhaps algorithms like Dreamer would significantly expand the scope of this work?
-  I request author to provide more discussion on MBPO-MAPER (Oh et. al) providing similarities and differences in the approaches.
- The theoretical results provide a bound on the difference between the policy's performance on learned model and the performance on the true model. How does that translate to actual policy improvement or convergence to optimal policy?

**Limitations:**

Authors discuss the limitation of their approach and suggest addressing them as future work.

**Strengths And Weaknesses:**

Strengths:
- This paper identifies a key challenge of model learning mismatch, experienced by model-based reinforcement learning algorithms.
- Their approach can be adapted with minimal change in the existing MBRL algorithm.
- The theoretical results provide finite time bounds on the performance difference of the policy for the true model and the learned model.
- The empirical performance clearly highlights the improved stability and, consequently, the improved sample complexity. Another interesting observation from their results is that the PMER does not degrade the global model error.
- The presentation of the work is very neat.


Weaknesses:
- Limited Scope of the Work: The empirical experiments build on ADMPO (also MBPO) and provide the PMER + ADMPO algorithm, and the experiments do show improved stability and better sample complexity. Algorithms such as Dreamer are more widely used in literature.  The paper's choice of building on ADMPO instead of more widely used algorithms, such as Dreamer (perhaps more challenging as well), significantly limits the scope of this work.

- The paper does not discuss a comparison with the Model augmented prioritized experience replay algorithm (Oh et al.). Oh et al. utilize model loss and do not solely rely on the TD error. They proposed an MBPO variant, MBPO-MAPER, and showed that it significantly improves the performance compared to MBPO. An empirical comparison with MBPO-MAPER, at least a detailed discussion of highlighting the differences and similarities, is essential.

---

> ### Author Rebuttal · Authors · 2026-03-31
>
> Thanks for your thoughtful assessment and encouraging comments. We are especially glad that you found the presentation clear and well organized, and that you recognized the motivation, simplicity, theoretical support, and empirical stability gains of PMER.
> Your main concerns appear to be the choice of ADMPO, the relation to MaPER-MBPO, and the practical implications of the theoretical bound. We address each point below and provide additional results in the linked file. The file is available [here](https://anonymous.4open.science/api/repo/PMER/file/1.pdf).
>
>
> > **Q1: Why ADMPO as a backbone rather than Dreamer?**
>
> We choose ADMPO as the main backbone because it is a representative replay-based, Dyna-style MBRL method and provides a clean testbed for studying the core mismatch studied in our paper. In ADMPO, the dynamics model is explicitly trained on replay data collected from a mixture of historical policies, and is later queried to improve the current policy. This directly matches the setting of our analysis.
>
> We do not use Dreamer as the main backbone because Dreamer-style methods introduce additional components, such as latent representation learning and latent-space imagination, which would substantially broaden the empirical scope and make it harder to isolate whether the gains come from reducing dynamics-model learning mismatch itself. Since the main goal of this paper is to identify, analyze, and address this mismatch in replay-based Dyna-style MBRL, ADMPO is the more appropriate primary testbed. A more detailed discussion of this design choice is provided in Appendix E of our paper.
>
> > **Q2: Generality and evaluation on more challenging domains.**
>
> Motivated by your feedback, we additionally evaluate PMER with a Dreamer-v3 backbone on Meta-World manipulation tasks. The PMER-enhanced variant shows improved learning efficiency and success rate across seeds, providing supporting evidence that PMER is not limited to the ADMPO/MuJoCo setting.
>
> The corresponding results are shown in Figure 9 of the [linked file](https://anonymous.4open.science/api/repo/PMER/file/1.pdf).
>
> We emphasize, however, that this experiment is included to demonstrate generality across different model-based RL paradigms, whereas the primary focus of this paper is on the mismatch in replay-based Dyna-style MBRL.
>
>
> > **Q3: MaPER-MBPO vs. PMER.**
>
> We have now completed an empirical comparison with MaPER-MBPO; the results are shown in Figure 7 of the linked file. For fairness, we report results on the four environments originally evaluated in MaPER. PMER shows stronger training stability than MaPER-MBPO, with smaller fluctuations, faster convergence, and better sample efficiency.
>
> Conceptually both methods use model-related signals to prioritize training samples in order to improve data efficiency in MBRL.
> However, they differ in both where prioritization is applied and what it is intended to improve.
> MaPER is a replay-prioritization method that augments PER with model-related auxiliary signals to improve Q-value learning; specifically, its priority combines TD error, reward prediction error, and transition prediction error using adaptive coefficients. In contrast, PMER applies prioritization only to dynamics model training, using a single dynamics-prediction-error signal motivated by the mismatch between the historical training distribution and the current-policy rollout distribution.
>
> Therefore, while both methods use model-related signals, MaPER mainly improves replay for value learning, whereas PMER directly targets dynamics model learning mismatch and rollout stability. PMER is also simpler to integrate, requiring only minimal changes to existing MBRL algorithms.
>
>
> > **Q4: Why does bounding the performance gap help improve MBRL?**
>
> Our theorem is not meant to establish convergence to the optimal policy directly. Rather, it clarifies why reducing the gap between policy performance in the learned model and in the true environment improves the reliability of model-based policy optimization. In Dyna-style MBRL, policy updates are computed using model rollouts, so the optimization is effectively performed with respect to $J_{\\hat{T}_\\phi}(\\pi_n)$.
>
> When the gap $|J_T(\\pi_n)-J_{\\hat{T}_\\phi}(\\pi_n)|$ is smaller, the learned-model objective is a more faithful surrogate for the true-environment objective, making policy updates less prone to exploiting model bias and instability. PMER is designed to reduce this mismatch by lowering policy-relevant local model error, thereby better aligning model-based improvement with real-environment improvement. This is also consistent with our empirical results, where PMER reduces oscillation and improves training stability.
>
> ---
> Overall, we sincerely thank you for your encouraging comments and insightful suggestions. Your questions about Dreamer and MaPER-MBPO led us to broaden our empirical validation. We hope these additions help clarify our contribution and address your concerns.

---

> > ### Author Rebuttal · Reviewer_Phft · 2026-04-04
> >
> > Thank you for addressing the questions in the rebuttal. The addition of the positive DreamerV3 results is an improvement that strengthens the paper's empirical claims. I will update my final recommendation to reflect this improvement.
> > I recommend authors provide more DreamerV3 experiments in the revised manuscript.

---

> > > ### Author Response · Authors · 2026-04-04
> > >
> > > Thank you for your reply and additional suggestion. We will incorporate this into account in the final version. Overall, we sincerely appreciate your careful review and constructive feedback, which helped us improve the paper!

---

### Official Review · Reviewer_nccN · 2026-03-13

**Soundness:** 3
**Presentation:** 3
**Significance:** 3
**Originality:** 2
**Overall Recommendation:** 4
**Confidence:** 3

**Summary:**

This paper identifies dynamics model learning mismatch as a central cause of instability in Dyna-style model-based reinforcement learning and proposes Prioritized Model Experience Replay (PMER), a lightweight replay mechanism that prioritizes high-error transitions during dynamics model training. The paper makes three core contributions: (i) A finite-horizon performance bound decomposing the policy performance gap into global model error, policy-induced distribution shift, and historical policy mixture effects; (ii) The PMER algorithm that uses prediction error as a proxy for policy mismatch without explicit policy distance estimation; and (iii) empirical demonstration on MuJoCo benchmarks showing improved stability and sample efficiency.

**Compliance With Llm Reviewing Policy:**

Affirmed.

**Final Justification:**

Concerns resolved, overall recommendation rasied from 3 to 4

**Key Questions For Authors:**

Assumption A.1's sensitivity coefficient $\delta_{T_ \phi}$: The assumption introduces a sensitivity coefficient that is only stated to be $\ge 0$, with no upper bound or typical values provided. The tightness of the final bound depends critically on the magnitude of $\delta_{T_ \phi}$, yet no empirical estimates are given.

TV contraction assumption in proof: The proof assumes "the dynamics model induces a TV contraction with constant $L_s$, but neural network dynamics models can have arbitrary Lipschitz constants depending on architecture and training. No empirical verification is provided that the trained models satisfy this contraction property.

Besides, the evaluation does not include:

Variance reporting across seeds (figures show mean ± SE but no statistical tests)
Sensitivity analysis on key hyperparameters beyond ρ (e.g., β, λ)

**Limitations:**

The theorem states the bound holds "under a short rollout horizon k" but never specifies what constitutes "short" or provides the range of k values for which the bound remains valid. Without this, readers cannot determine whether their intended rollout horizon falls within the valid regime.

Lacking tests on more challenging domains (e.g., human-level control tasks, sparse reward settings)

**Strengths And Weaknesses:**

# Strengths:
The paper addresses an important and practical problem in MBRL training instability. The theoretical analysis provides useful intuition about why minimizing global error alone may be insufficient. PMER is elegantly simple and integrates seamlessly into existing MBRL frameworks without architectural changes. The empirical evaluation includes multiple baselines and ablation studies demonstrating consistent gains across environments.

# Weaknesses:
Several critical assumptions in the theoretical analysis lack empirical verification or bounds. Theorem 3.1's "short rollout horizon k" is undefined, Assumption A.1's sensitivity coefficient $\delta_{T_\phi}$ is unbounded, and the proof's TV contraction assumption ($L_s$ < 1) is not verified for trained neural network models. The core proxy assumption (prediction error correlates with policy mismatch) is deferred to appendix without main-text quantitative summary. The claim about PMER improving the consistency factor $L_{s^(k)}$ lacks direct measurement. Writing clarity issues include undefined notation (time index t in global error definition) and ambiguous normalization procedures in Eq. 7.

---

> ### Author Rebuttal · Authors · 2026-03-31
>
> Thanks for your valuable feedback! Responses are as follows. The linked supplementary file is available [here](https://anonymous.4open.science/api/repo/PMER/file/1.pdf).
>
> > **Q1: $k$ - What is the short rollout horizon $k$?**
>
> The notion of a short rollout horizon $k$ follows standard practice in MBPO-style methods (Janner et al., 2019), where short model rollouts are used to control compounding error, and has been widely adopted in subsequent MBRL work. In our implementation, $k$ is explicitly specified in Appendix E (model rollout schedule). We will clarify this explicitly in the revision.
>
> > **Q2: $\\delta_{\\hat{T}_\\phi}$ - sensitivity coefficient.**
>
> We provide empirical evidence in Figure 5 of the linked file showing that this sensitivity term remains well-controlled in practice. Specifically, we train a predictive model on the state distribution induced by $\\pi_{\\mathrm{mix},n}$, and then evaluate it without retraining on newly constructed policies $\\pi$ with varying KL divergence.
> We estimate $d(\\epsilon_{m'})/d(\\epsilon_\\pi)$ by numerical differentiation. The sensitivity decreases from about $10^{-2}$-$10^{-1}$ early in training to around $10^{-4}$ later. This matches the training dynamics: as shown in Figure 6 of our paper, the local and global errors overlap near convergence, indicating reduced sensitivity to policy shift.
>
> > **Q3: $L_s^{(k)}(\\hat{T}_\\phi)$ - contraction property / consistency coefficient estimate.**
>
> We agree that strict global verification of TV contraction for learned dynamics is challenging. In our analysis, $L_s^{(k)}(\\hat{T}_\\phi)$ is a finite-horizon TV-Lipschitz coefficient under short rollouts, rather than a claim of global contraction. Under small-$k$ Dyna-style rollouts, it is empirically well controlled.
>
> Empirically (see Fig. 6 in the linked file), the coefficient remains below $1$ throughout training on Hopper. PMER achieves lower values than ADMPO ($\\approx 0.2$ vs. $\\approx 0.4$ at convergence), indicating improved finite-horizon multi-step consistency. This is consistent with the mechanism in Fig. 1 and the error trends in Figs. 6-7 of our paper.
>
> > **Q4: $t$ — What is the time index $t$?**
>
> The time index $t$ denotes the step within a finite-horizon rollout, rather than the training iteration index. Accordingly, $\\rho^{\\pi_{\\mathrm{mix},n},t}$ denotes the state-action distribution at rollout step $t$ induced by the mixture policy $\\pi_{\\mathrm{mix},n}$ over $n$ historical policies.
>
> > **Q5: Normalization procedures in Eq. 7.**
>
> No additional normalization is applied to the priority score in Eq. 7 itself. The normalization may refer to standardizing the inputs to the dynamics model. Specifically, we compute running means and standard deviations for the state and action, and normalize them before input to the model (see `def set_mu_std(*)` in our implementation). This is commonly used in MBRL for stabilizing training and is independent of our prioritization.
>
> > **Q6: Variance reporting.**
>
> MBRL training is inherently unstable, so convergence-based scalar summaries can be noisy. Consistent with prior MBRL works, we therefore regard the learning curves with seed-wise uncertainty as the primary evidence.
> We additionally report convergence-time variance (defined as the step at which performance reaches 90\% of the asymptotic return).
> PMER shows consistently lower variance than ADMPO:
> Hopper (4.70 vs. 98.50), Humanoid (30.70 vs. 62.50), Ant (70.67 vs. 100.00), Walker (25.33 vs. 76.33), HalfCheetah (4.00 vs. 16.33).
>
> > **Q7: Ablation of $\\lambda$ and $\\beta$.**
>
> We refer to Figures 2-4 in the linked file. For $\\lambda$, removing decay leads to clear underperformance because stale high-error samples persist in the buffer, indicating that dynamic decay is necessary. By contrast, overly strong decay ($0.9$) suppresses high-error samples too aggressively, making PMER less effective at correcting local mismatch.
>
> For $\\beta$, the best convergence is observed around $\\beta \\in [2,3]$ on both Hopper and Humanoid. Please also see our response to Reviewer RMcy for a more detailed discussion of the $\\beta$ ablation.
>
> > **Q8: More challenging domains.**
>
> Our main experiments follow prior evaluation protocols in MBRL and use the same MuJoCo benchmark suite as in prior work, including MBPO, PDML, and ADMPO, enabling fair comparison under established settings.
> We agree that evaluation on more challenging domains would further strengthen the paper. In response, we additionally evaluate PMER in a Dreamer-v3 framework on more challenging tasks, where it shows consistent improvements (Figure 9, linked file). Please also see our response to Reviewer Phft for further details.
>
> ---
> We appreciate your comments, which helped us make the paper more complete and better clarify its scope and assumptions. We hope the added analyses and clarifications make the assumptions, empirical support, and intended scope of the paper much clearer!

---

> > ### Author Rebuttal · Reviewer_nccN · 2026-04-02
> >
> > Thanks for the extensive results and the thoughtful reply. What I'm really asking about k is its value among all environments and the analysis of its impact. I checked Appendix E but only found a sentence like "k=1 among all MBPO environments". Is that true? Have you tried other values?

---

> > > ### Author Response · Authors · 2026-04-03
> > >
> > > Thanks for your prompt feedback!
> > >
> > >  We apologize for the typo in our rebuttal. The correct reference is "Appendix F (model rollout schedule)" in Table 2, not "Appendix E (model rollout schedule)". Specifically, the rollout-horizon $k$ settings across environments are:
> > >
> > > | Environment | HalfCheetah | Hopper | Walker2d | Ant | Humanoid |
> > > |---|---|---|---|---|---|
> > > | Model rollout horizon $k$ schedule  | 1→1 over 0→200k | 1→15 over 0→50k | 1→10 over 0→100k | 1→5 over 10k→100k | 1→10 over 10k→100k |
> > >
> > > x → y over a → b denotes a thresholded linear increasing schedule, i.e. the model rollouts horizon at environment step $t$ is calculated by $f(t)=\\min\\left(\\max\\left(x+\\frac{t-a}{b-a}(y-x),\\,x\\right),\\,y\\right).$
> > >
> > > Here “$1 \\rightarrow 5$ over $10k \\rightarrow 100k$” means that $k$ is gradually increased from 1 to 5 as training proceeds from 10k to 100k environment steps (otherwise $k=1$). This "short rollout horizon" setting was proposed in MBPO (Janner et al., 2019) and was adopted by ADMPO, as shown in their Appendix D.2, as well as by later MBPO-style works.
> > >
> > > > **What is the impact of $k$ ?**
> > >
> > > The rollout horizon $k$ is how many consecutive imagined steps the learned model generates. The resulting imagined transitions are added to the model buffer and used for policy training.
> > > For example, if $k=3$, the model generates a 3-step imagined trajectory starting from a real state, and these synthetic transitions are then used together with real transitions to update the policy.
> > >
> > > As shown above, $k$ starts from 1 and increases only to a small value, which is exactly what we mean by a short-horizon rollout. If $k$ is too large, compounding error grows quickly, making the imagined data less reliable and more likely to cause model exploitation and unstable learning. On the other hand, if $k$ is too small, model rollouts provide only limited benefit for policy learning.
> > >
> > > We kept exactly the same rollout settings as the backbone method, so that the gains of PMER cannot be attributed to different rollout-horizon choices. We therefore did not evaluate other rollout horizons.
> > >
> > > > **But what does the “in MBPO, $k=1$” mean in the discussion around Eq. (7)?**
> > >
> > > This refers to a different quantity from the rollout-horizon $k$ in Eq. (2) and Theorem 3.1.
> > >
> > > In Eq. (7), $k$ denotes the prediction span used to compute the priority weight, i.e., the length of the any-step / multi-step target, rather than the model rollout horizon.
> > > To avoid this ambiguity, we will revise the notation by keeping $k$ for the rollout horizon and using $k_p$ for the prediction span in Eq. (7).
> > >
> > > Concretely, MBPO uses one-step prediction only when training the dynamics model, i.e.,
> > > $\\mu_{\\theta}(s_t, a_t) \\rightarrow (s_{t+1}-s_t, r_t).$
> > > By contrast, ADMPO uses any-step prediction and samples $k_p$ from $1$ to $m$, where $m$ is the maximum back-tracking length, i.e.,
> > > $\\mu_{\\theta}(s_t, a_{t:t+k_p-1}) \\rightarrow (s_{t+k_p}-s_t, r_{t+k_p-1}).$
> > >
> > > Accordingly, Eq. (7) should be written as
> > > $\\psi_i^n=\\left\\|\\mu_\\theta\\left(s_t^{(i)},a_{t:t+k_p-1}^{(i)}\\right)-\\left(s_{t+k_p}^{(i)}-s_t^{(i)},r_{t+k_p-1}^{(i)}\\right)\\right\\|_2^2.$
> > >
> > > For MBPO, since the model is trained with one-step prediction only, this quantity is fixed to $k_p=1$, which reduces Eq. (7) to
> > > $\\psi_i^n=\\left\\|\\mu_\\theta\\left(s_t^{(i)},a_{t}^{(i)}\\right)-\\left(s_{t+1}^{(i)}-s_t^{(i)},r_{t}^{(i)}\\right)\\right\\|_2^2.$ This is why "in all MBPO experiments, $k_p=1$" in Appendix E.
> > >
> > > For ADMPO, the maximum range $m$ follows the original ADMPO setting. For fairness, we did not modify this setting when applying PMER. Specifically, we use $m=5$ in Hopper (i.e., $k_p \\in [1,5]$) and $m=2$ in the other environments, consistent with both our implementation and the original ADMPO setup. We will clarify it explicitly in the revision.
> > >
> > > ---
> > > We thank you again for pointing out this ambiguity. We will address it explicitly in the revision by:
> > >
> > > - (1) stating the rollout horizon $k$ used in our experiments explicitly in the manuscript;
> > > - (2) separating the rollout horizon $k$ from the prediction span $k_p$ in Eq. (7) using different notation;
> > > - (3) clarifying the maximum back-tracking length $m$ and the prediction span $k_p$.
> > >
> > > If you feel that the rebuttal has sufficiently resolved your concerns, we would be very grateful if you could consider updating your score accordingly.

---

### Official Review · Reviewer_RMcy · 2026-03-13

**Soundness:** 3
**Presentation:** 3
**Significance:** 3
**Originality:** 3
**Overall Recommendation:** 5
**Confidence:** 4

**Summary:**

The paper identifies in the context of model-based RL a mismatch between the historical data that is used to train a dynamics model and the data region where the dynamics model is queried on afterwards. Thus, the paper differentiates local and global model error and proposes a lightweight mechanism to prioritise highly relevant data in the dynamic model training. The mechanism is then demonstrated on an ADMPO backbone in the paper (wich some MBPO backbone experiments in the appendix) and compared to baselines on the MuJoCo suite.

**Compliance With Llm Reviewing Policy:**

Affirmed.

**Final Justification:**

I thank the authors for their further comments. Since my concerns have been addressed, I updated my overall score from 4 to 5.

**Key Questions For Authors:**

(1) Can you go in more detail about your observation "that transitions with large dynamics model prediction error naturally correspond to policy-induced distribution shift"? The paper states so in the introduction, but does not provide further details. Maybe here could be a good spot to talk about to what kind of problems PMER is limited to.

(2) Can you elaborate: how does PMER work under stochastic environment dynamics? (Not necessarily highly stochastic environment dynamics.)

(3) For the \beta sampling, how is the statement "In practice, we find \beta \in [2,3] to provide a good trade-off" motivated? What happens outside this interval?

**Limitations:**

yes

**Strengths And Weaknesses:**

Weaknesses

(a) Limitations of PMER addressed at the end of the conclusion section seem to be vital limitations to the overall presented methodology. Thus it would be important to address and discuss them earlier in a broader way.

(b) Benefits of PMER seem to be incremental with respect to significance and originality.

(c) Minor: There are bits in the paper that lack motivation and context, see also questions.

(d) Minor: The manuscripts repeats itself frequently to highlight the difference between local and global error.


Strengths

(e) The paper seems overall sound.

(f) The paper motivates and presents the problem well with helpful plots. Section 5.4 and 5.5 with Figures 6 und 7 are very helpful.

---

> ### Author Rebuttal · Authors · 2026-03-31
>
> Thanks for the careful reading and positive assessment. We are encouraged that you find the paper technically sound and the presentation helpful, especially Sections 5.4-5.5 and Figures 6-7. In response to your main concerns regarding the scope, significance, and empirical motivation of PMER, we provide additional clarifications here and include the corresponding experimental results in the [linked supplementary file](https://anonymous.4open.science/api/repo/PMER/file/1.pdf).
>
> > **Q1: Why does model prediction error naturally correspond to policy-induced distribution shift?**
>
> Policy updates induce distribution shift by pushing the rollout distribution toward regions more relevant to the current policy. Compared with the historical-policy mixture in the replay buffer, these regions are often less well covered, more underfitted, and thus associated with larger local prediction error. Therefore, prediction error corresponds to policy-induced distribution shift because the shift is revealed precisely through elevated error in these undercovered regions. We support this empirically in Appendix D. Building on this mechanism, Figure 6 further shows that PMER mitigates the learning mismatch by better aligning the reduction of local and global errors.
>
> We also include an additional experiment in Figure 1 of the linked supplementary file. It shows that high-error samples are concentrated in regions closer to the current policy, while samples farther from the current policy often exhibit lower error because they have been better covered during model training. This supports our intuition that model prediction error naturally highlights underfitted, policy-relevant regions. Thus, prioritizing high-error transitions can implicitly emphasize samples relevant to the current policy, without explicitly computing policy-distance weights.
>
> > **Q2: What kind of problems is PMER limited to / How does PMER behave under stochastic dynamics?**
>
> PMER is less effective in highly stochastic environments, where prediction error is dominated by intrinsic randomness rather than model bias. For example, in stochastic grid worlds with random slip or wind dynamics, large prediction errors may arise even in well-covered regions. In such cases, high-error transitions no longer reliably reflect policy-induced distribution shift, making prioritization less informative. We agree that this limitation should be stated more clearly, and in the revised version we will discuss it earlier in the paper.
>
> > **Q3: PMER seems incremental.**
>
> While PMER is simple algorithmically, we believe the contribution is not merely incremental. The paper identifies and formalizes dynamics model learning mismatch, and introduces a lightweight replay mechanism that directly targets this issue without explicit policy-distance estimation. By using a dynamically updated buffer instead of reweighting the full replay set, PMER also reduces computational overhead and is meaningfully different from prior replay-based designs.
>
>
>
> > **Q4: $\\beta$ settings.**
>
> Thanks for pointing out this detail. We now provide the $\\beta$ ablation in Figures 2 and 3 of the supplementary file. Specifically, we evaluate $\\beta \\in \\{1,2,3,4\\}$. PMER works well over a broad range, although not all settings are equally effective. We observe the best convergence around $\\beta \\in \\{2,3\\}$ on both Hopper and Humanoid, while values outside this range can cause mild performance degradation on some tasks.
>
> When $\\beta$ is small, sampling is closer to uniform over the prioritized buffer, so high-error transitions are not emphasized enough, limiting the correction of local mismatch. When $\\beta$ is large, sampling concentrates more heavily on the highest-error transitions. While this can better target mismatch-relevant samples, it also reduces sample diversity, which may slow convergence and harm stability. We also provide the $\\lambda$ ablation in Figure 4.
>
> ---
> We hope these additional results and clarifications help address your concerns and better convey the intended contribution of the paper.

---

> > ### Author Rebuttal · Reviewer_RMcy · 2026-04-03
> >
> > Thank you for your elaborations. My concerns have been addressed and I'm considering of adjusting my score.

---

> > > ### Author Response · Authors · 2026-04-03
> > >
> > > Thank you very much for your thoughtful response! We are glad that our rebuttal addressed your concerns, and we sincerely appreciate your consideration of our work and your constructive feedback!

---

### Decision · Program_Chairs · 2026-04-30

**Decision:**

Accept (regular)

**Comment:**

The paper has been received unanimously positively.

The rebuttal was very successful, as was the discussion, resulting in significantly more positive ratings from the reviewers.

Overall, a convincing, solid piece of work of high quality.